# SurVIndel2: improving copy number variant calling from next-generation sequencing using hidden split reads

Ramesh Rajaby [1,2,3,6] & Wing-Kin Sung [1,2,3,4,5] ✉

Deletions and tandem duplications (commonly called CNVs) represent the majority of structural variations in a human genome. They can be identified using short reads, but because they frequently occur in repetitive regions, existing methods fail to detect most of them. This is because CNVs in repetitive regions often do not produce the evidence needed by existing short reads-based callers (split reads, discordant pairs or read depth change). Here, we introduce a new CNV short reads-based caller named SurVIndel2. SurVindel2 builds on statistical techniques we previously developed, but also employs a novel type of evidence, hidden split reads, that can uncover many CNVs missed by existing algorithms. We use public benchmarks to show that SurVIndel2 outperforms other popular callers, both on human and non-human datasets. Then, we demonstrate the practical utility of the method by generating a catalogue of CNVs for the 1000 Genomes Project that contains hundreds of thousands of CNVs missing from the most recent public catalogue. We also show that SurVIndel2 is able to complement small indels predicted by Google DeepVariant, and the two software used in tandem produce a remarkably complete catalogue of variants in an individual. Finally, we characterise how the limitations of current sequencing technologies contribute significantly to the missing CNVs.

Structural Variations (SVs) are mutations in a genome that involve 50 nucleotides or more. Although they are less frequent than point mutations, SVs account for more heritable differences in populations due to their large scale and play a major role in many genetic diseases[1]. Deletions and tandem duplications are two important classes of SVs. Deletions occur when a contiguous sequence of bases is lost, while tandem duplications occur when a source sequence of DNA is duplicated in place; they are called unbalanced SVs because they alter the copy number of the genome. A recent study of the SVs in 14,891 human genomes[2] identified deletions as the most frequent

SV and tandem duplications as the third most common, after insertions. However, the SVs were detected using short reads, and existing SV callers missed most duplications, as demonstrated in the following sections. When detecting SVs using high-quality long reads, tandem duplications are as common as deletions (Section 2.3). Aside from being an important source of polymorphism, deletions and tandem duplications are implicated in many genetic diseases, such as Smith-Magenis syndrome, Potocki-Lupski syndrome and Williams-Beuren syndrome[1]. We previously defined deletions and tandem duplications as *local copy number variants* (local CNVs)[3]. In this

[1]Department of Chemical Pathology, The Chinese University of Hong Kong, Hong Kong, China. [2]Hong Kong Genome Institute, Hong Kong Science Park, Shatin, Hong Kong, China. [3]A*STAR Genome Institute of Singapore, Singapore, Singapore. [4]JC STEM Laboratory of Computational Genomics, Li Ka Shing Institute of Health Sciences, The Chinese University of Hong Kong, Hong Kong, China. [5]School of Computing, National University of Singapore, Singapore, Singapore. [6]Present address: Shibuya Lab, Division of Medical Data Informatics, Human Genome Center, University of Tokyo, Tokyo, Japan. ✉e-mail: kwksung@cuhk.edu.hk

manuscript, we will often refer to them simply as CNV for convenience.

Nowadays, CNVs can be accurately detected using long-read sequencing technologies, like PacBio HiFi reads and ONT nanopore reads. However, these technologies are still expensive and this limits their use in large-scale population studies. A less costly solution is to use short-read sequencing technologies such as Illumina, which is often used for sequencing large populations[2,4,5]. Short-read datasets are rapidly increasing in number, with studies on tens of thousands of genomes already published. As the Hong Kong Genome Project, Singapore SG100K, the European '1 + Million Genomes' Initiative, and the All of Us Research Project (among others) continue to generate large amounts of sequencing data, short-read whole genome sequencing will remain at the forefront of genomics for the foreseeable future.

However, detecting deletions and especially tandem duplications from short reads is very challenging. Despite a large number of callers published in the last decade, the sensitivity of even the best methods is still low. Manta, a widely used SV caller employed in many large-scale population studies[2,5], as well as the best-performing tool according to recent benchmarking efforts[6,7], detects around 40% of the existing deletions and less than 10% of the existing tandem duplications when tested on a recent benchmark dataset (see Section 2.3).

In a previous study[3], we found that most of the CNVs missed by existing callers were located in tandem repetitive regions. This is because CNVs in tandem repetitive regions often do not generate split reads; this result was confirmed in this study (Section 2.1). Furthermore, shorter CNVs generate few or no discordant pairs, and read depth methods are known to only work for large CNVs[8]. Thus, naively counting the number of discordant pairs and split reads will miss many CNVs in repetitive regions. Finally, we showed that the problem can be mitigated by considering the insert sizes of all the read pairs containing the putative CNVs and performing statistical tests to determine whether they deviate from expectations. Using this technique we developed SurVIndel[3], a CNV caller that improved the detection rate in repetitive regions over existing methods.

However, SurVIndel suffered from two problems. First, it generated many false positives. This is a serious issue when generating catalogues of CNVs for large populations. Second, because split reads are often unavailable in repetitive regions, many of the CNVs detected had imprecise boundaries. Subsequent steps in population studies, such as clustering of the CNVs across samples, genotyping and downstream analysis, all rely on and greatly benefit from precise breakpoints.

To address these issues, we have developed a new caller called SurVIndel2, which significantly reduces the number of false positives while improving the sensitivity of the original SurVIndel. In addition, SurVIndel2 generates precise breakpoints for most of the called CNVs. This new caller utilises split reads, discordant pairs and a novel type of evidence called hidden split reads to detect candidate CNVs. Hidden split reads allow us to identify the existence and the precise breakpoints of CNVs in repetitive regions, which leads to a more accurate and comprehensive set of candidate CNVs. SurVIndel2 then applies the statistical approach of SurVIndel, along with depth-based filters, to generate a set of confident CNVs.

This paper is organised as follows. In Section 2.1, we introduce the notion of hidden split reads, and we explain why they can improve CNV detection. Section 2.2 gives an overview of the algorithm of SurVIndel2. Sections 2.3 and 2.4 use publicly available datasets, both human and non-human, to compare the performance of SurVIndel2 to the best methods in the literature, according to a recent benchmark[7]. Then, in Section 2.5, we use SurVIndel2 to generate a dataset of deletions and duplications in the 1000 Genomes Project. We show that our dataset is significantly more complete than that produced by a state-of-the-art pipeline: it contains more than twice as many deletions and five times as many duplications, with high validation rates according to

long reads. In Section 2.6, we illustrate that SurVIndel2 is also able to effectively call CNVs that are smaller than 50 bp, and it can be used to enhance callsets generated by state-of-the-art small variants callers such as DeepVariant[9]. Finally, in Section 2.7, we investigate why many CNVs are hard to detect, and we reach the surprising conclusion that it is not only due to the repetitive nature of the genome but mostly due to the sequencing platforms employed being unable to correctly sequence many regions of the genome.

SurVIndel2 is fully open source and available at https://github.com/kensung-lab/SurVIndel2. It is easy to run and only requires a BAM or a CRAM file and a reference genome, and it outputs a fully standard VCF file.

## Results

### Hidden split reads

Let $D$ be a deleted region in a sample, and let $L$ and $R$ be its left and right-flanking regions, respectively. $L$ and $R$ will be adjacent in the sample genome, but not in the reference genome. The concatenation of $L$ and $R$ (we refer to it as $LR$) is called the junction sequence of the deletion. When the junction sequence is a novel sequence, i.e., it is not present in the reference genome, the reads sequenced from it will be "split", meaning that a portion of the read will align to $L$ and the rest will align to $R$. Split reads are commonly used to detect deletions. Figure 1a shows an example of a deletion creating a split read. Similarly, tandem duplications may also produce split reads (Fig. 1b).

Some deletions will not produce split reads. One reason for this is that the junction sequence is not novel. For example, if a sufficiently long prefix of $D$ is homologous to a prefix of $R$, the deletion will not create split reads (Fig. 1c). Similarly when a suffix of $D$ is homologous to a suffix of $L$, no split reads will support the deletion (Fig. 1d). Tandem repeats are common examples of this.

We have a tandem repeat when a sequence (*repeat unit*) appears consecutively multiple times. Deletions and duplications of entire repeat units often do not create split reads[3]: because $L$, $R$ and $D$ are identical, the junction sequence $LR$ will be identical to $LD$ and to $DR$, so any read sequenced from $LR$ will fully align to either $LD$ or $DR$ (Supplementary Fig. 4a).

However, in practice, the copies in a tandem repeat may present some differences between each other. Suppose $D$ and $R$ are similar but not identical, and let $r$ be a read sequenced from $LR$. The algorithm that aligns $r$ to the reference genome has two choices:

- split $r$ between $L$ and $R$, obtaining two (partial) perfect alignments; or
- align $r$ fully to either $LD$ or $DR$, accepting mismatches or gaps in the alignment.

Which option the aligner chooses depends on many factors, including the mapping algorithm used, its parameters and how similar $L$, $D$ and $R$ are. The more different they are, the more likely option 1 will be chosen. When option 2 is chosen, we say that $R$ is a *hidden split read* (abbreviated as HSR, see Fig. 1e for an example). Similarly, tandem duplications may also produce hidden split reads (Supplementary Fig. 4b).

Next, we provide a formal definition of hidden split read. Given a genomic fragment $s$, let $score(s)$ be the score of the alignment of $s$ to the reference genome. Given a read $r[1..n]$, we say that $r$ is a hidden split read if there exists $i \in [1..n-1]$ such that $score(r) < score(r[1..i]) + score(r[i+1..n])$; in other words, if $r$ is allowed to split into two without penalty and the two parts can align independently, the score of the alignment will increase. $score(r[1..i]) + score(r[i+1..n]) - score(r)$ is the *HSR-score* of $r$. See Supplementary Fig. 5 for an example of HSR-score calculation.

In order to confirm that CNVs mostly delete or insert full repeat units and the importance of hidden split reads in detecting such CNVs, we use two publicly available human genomes: HG002 and HG00512. A

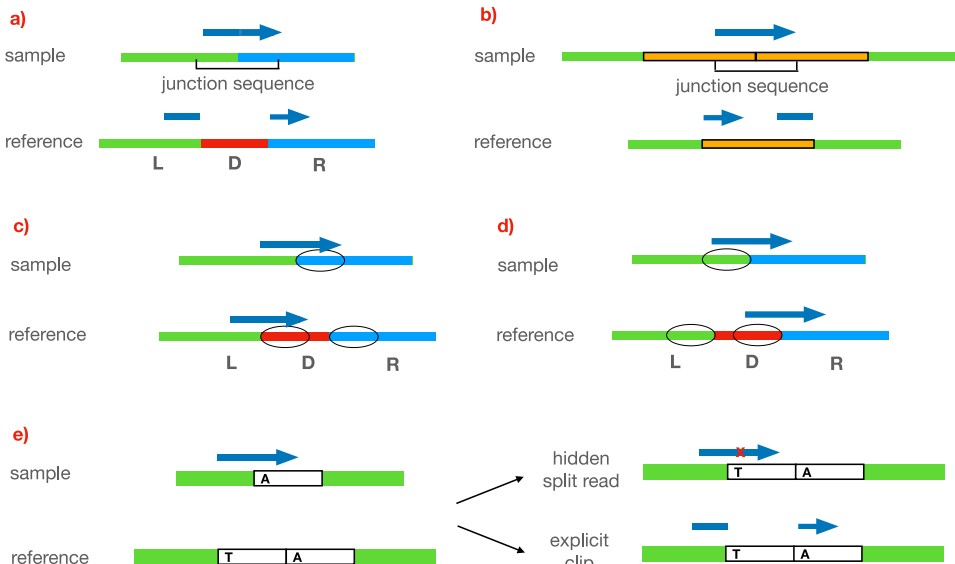

**Fig. 1 | Split read, and hidden split reads. a** The red segment (*D*) of the reference is deleted in the sample, causing the green (*L*) and the blue (*R*) regions to be adjacent. Reads sequenced partly from the *L* and partly from *R* may not fully align with the reference, causing the read to be clipped. **b** Similarly, tandem duplications may also create clipped reads. **c** When the prefixes (circled) of *D* and *R* are homologous, many reads sequenced across the breakpoint of the deletion will fully align to the reference, instead of being clipped. If the length of the prefixes is longer than the read length, no clipped reads will be produced. **d** Similarly, when the suffixes of *D* and *L* are homologous, little or no clipped reads will be produced. **e** Deletions of full copies in tandem repeats are a prime example of (**c**), which is why they often do not cause clipped reads. However, in practice, different copies of a tandem repeat are often not identical, and we may get hidden split reads. In this example, a sequence (white) is repeated twice in the reference, but only once in the sample. The two copies of the sequence are identical except for one position (the first copy has a T, and the second has an A). The single copy in the sample has an A, therefore we want to report that the first copy was deleted. Reads spanning both the green left-flanking region and the A character can be aligned to the reference in two ways: fully with one mismatch, or explicitly clipped. Aligners will normally choose the former. While this is usually a reasonable choice, in this case, it will hide the existence of the deletion. We call such reads hidden split reads.

detailed explanation of the data used and the experiments mentioned below is provided in Methods.

Around 64% of the deletions and 93% of the tandem duplications in HG002 (Fig. 2a) are located in tandem repetitive regions (identified by TRF[10], "Methods"). Fig. 2b demonstrates that, indeed, most CNVs delete or insert an integer number of repeat units. For this reason, the number of CNVs in tandem repeats supported by split reads is low. Only 25.2% of the deletions and 29.0% of the duplications in tandem repeats were supported by split reads, compared to 90.2% and 83.3%, respectively, for deletions and duplications not in tandem repeats (Fig. 2c, d). However, hidden split reads can help us uncover a significant number of CNVs not supported by split reads. Using both split reads and hidden split reads, the number of deletions in tandem repeats that we are potentially able to discover is more than double (25.2% using split reads, 58.9% using both split reads and hidden split reads, 2.3×). We obtain similar figures for duplications (29.0% using split reads, 63.7% using both split reads and hidden split reads, 2.2×). Supplementary Fig. 6 shows that the statistics are virtually identical for HG00512.

Although hidden split reads have the potential to identify a substantial portion of repetitive CNVs that are typically missed by traditional split-read methods, their use presents significant challenges. Fig. 2e shows the binned HSR-score distribution for hidden split reads that support CNVs in HG002. Most hidden split reads have low HSR scores. Fig. 2f shows that most CNVs detected by realigning hidden split reads are actually noise. Interestingly, reads with high HSR scores do not produce significantly fewer false positives in relative terms. Distinguishing hidden split reads that support existing CNVs is not trivial, and novel and sophisticated techniques need to be developed. In the remainder of this paper, we propose an algorithm that uses hidden split reads to detect deletions and duplications in a sample genome, and significantly outperforms the current state of the art.

## Overview of the method

SurVIndel2 requires (1) a reference genome (in FASTA format) and (2) a BAM or CRAM alignment file containing paired reads sequenced from a single sample aligned to the reference genome. It outputs a VCF file containing a list of predicted deletions and tandem duplications. Figure 3 shows a high-level overview of the full algorithm. Detailed explanations and examples for individual steps are provided in Supplementary Methods.

As a first step, SurVIndel2 iterates through the alignment file and gathers the evidence that will be later used to determine the candidate CNVs: split reads, hidden split reads and discordant pairs. Split reads, and hidden split reads were described in Section 2.1. SurVIndel2 considers potential hidden split reads to be all the reads that are not clipped and present at least three differences with the reference genome (defined as a number of mismatches plus the number of indels). Discordant read pairs are pairs of reads that map inconsistently to the library preparation parameters. A deletion may produce pairs where the two reads map very far apart (Supplementary Fig. 17a). Let $\mu$ and $\sigma$ be the mean and the standard deviation of the distribution of insert sizes for the library: we define as discordant any pair that has insert size greater than $\mu + 3\sigma$ (we will refer to this quantity as $max_{IS}$ throughout the paper). On the other hand, a duplication may produce pairs that have the wrong relative orientation, i.e., the forward-facing read is mapped downstream of the reverse-complemented read (Supplementary Fig. 17b). We define such pairs as discordant.

Given the split reads, hidden split reads, and discordant pairs, candidate CNVs are generated by two different modules: the *split module* uses split reads and hidden split reads, while the *discordant module* uses discordant pairs. Then, we calculate a set of features for each putative CNV, such as the number of supporting split and hidden split reads, their mapping quality, the number of discordant and concordant pairs, alignment score of the junction sequence when

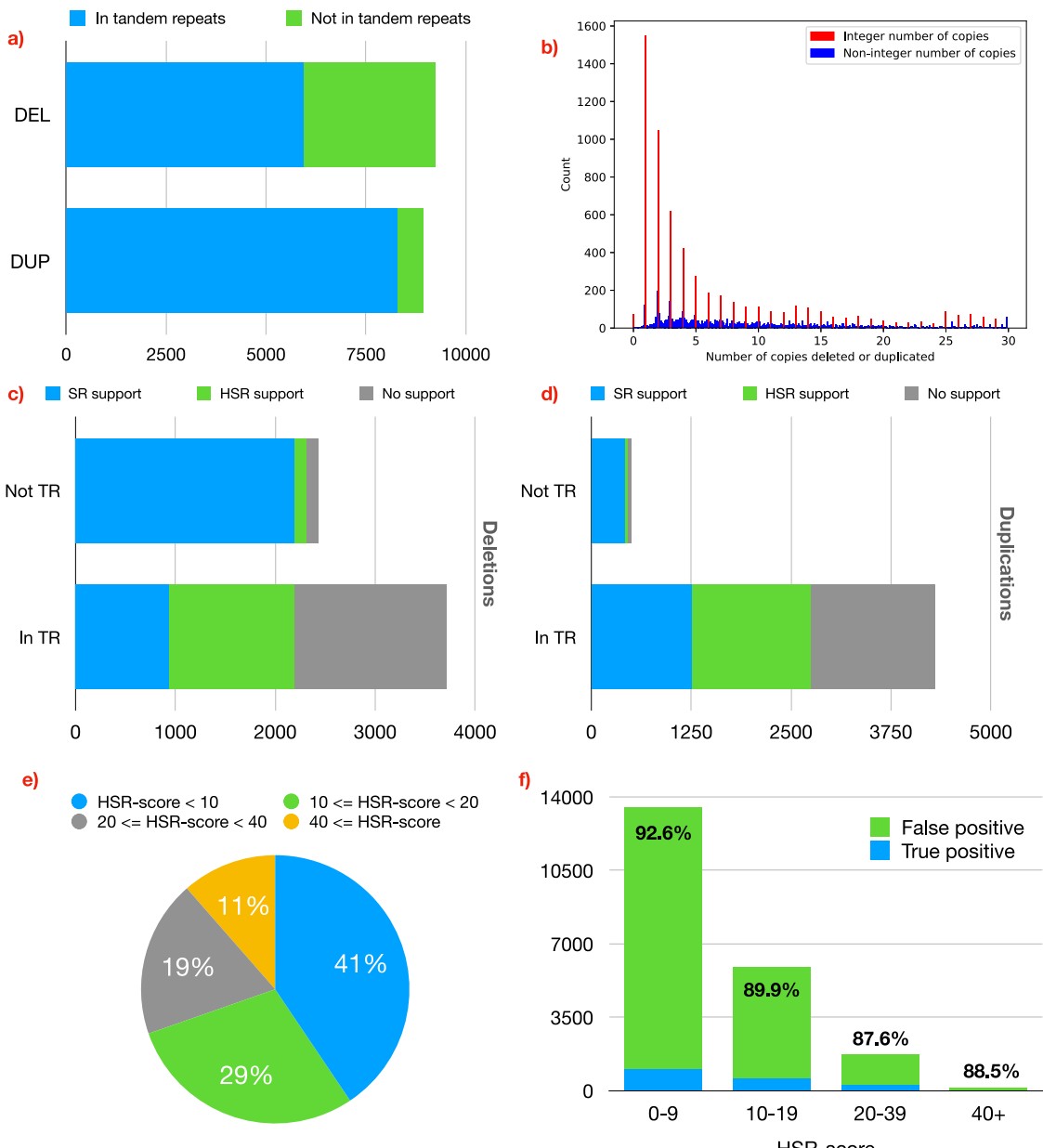

**Fig. 2 | The impact of copy number variants (CNV) deleting or duplicating full repeat units and the importance of hidden split reads in detecting such CNVs, studied on HG002. a** We consider a CNV to be contained in a tandem repeat if at least 90% of the CNV is covered by the repetitive region. Nearly two-thirds of the deletions and more than 93% of the tandem duplications in this dataset are contained in a tandem repeat. **b** For every CNV that overlaps with a tandem repeat, we calculated the number of repeat units that are deleted or inserted, by dividing the length of the CNV by the length of the repeat unit. Bars representing an integer number of copies (i.e., the length of the CNV is a multiple of the length of the repeat unit) are coloured red. Most CNVs add or delete an integer number of repeat units. **c**, **d** Number of deletions (**c**) and tandem duplications (**d**) inside and outside of tandem repeats that are supported by split reads (SR), hidden split reads (HSR), and neither. Hidden split reads have the potential to recover a significant number of deletions and duplications missed by regular split reads. **e** HSR scores of the hidden split reads supporting the existing CNVs in HG002. Most reads have a low (< 20) HSR score. **f** We found all hidden split reads in 10,000 randomly sampled repetitive regions, and for each, we tried to predict a CNV by finding the optimal split alignment. Most CNVs (> 91%) detected this way were false positives. This shows that using hidden split reads to detect CNVs is challenging, and effective filters must be employed to distinguish real from false CNVs.

realigned to the reference, depth of the CNV and flanking regions, and others (Fig. 3h). In addition, we use the distribution of the insert sizes of the read pairs around the CNV to calculate a confidence interval for its size, as well as a *p*-value estimating the probability that the CNV is a false positive. Such features were first introduced by SurVIndel[3]. Finally, all of the features are used in the filtering step (Fig. 3g), which aims at removing most false positives, while retaining the correct CNVs. In order to do so, it uses a pre-trained random forest. If a pre-trained forest is not available, a set of fixed hard filters is used. Below, we explain how the split and the discordant modules find candidate CNVs.

The split module first generates consensus sequences by clustering left-clipped reads and hidden split reads into left-clipped consensus sequences and right-clipped reads and hidden split reads into right-clipped sequences (Fig. 3a, b). The consensus sequences are partitioned into left and right-clipped. Consensus sequences in each group potentially represent either the left or the right breakpoint of a CNV: furthermore, using the mates of the clustered reads, we can

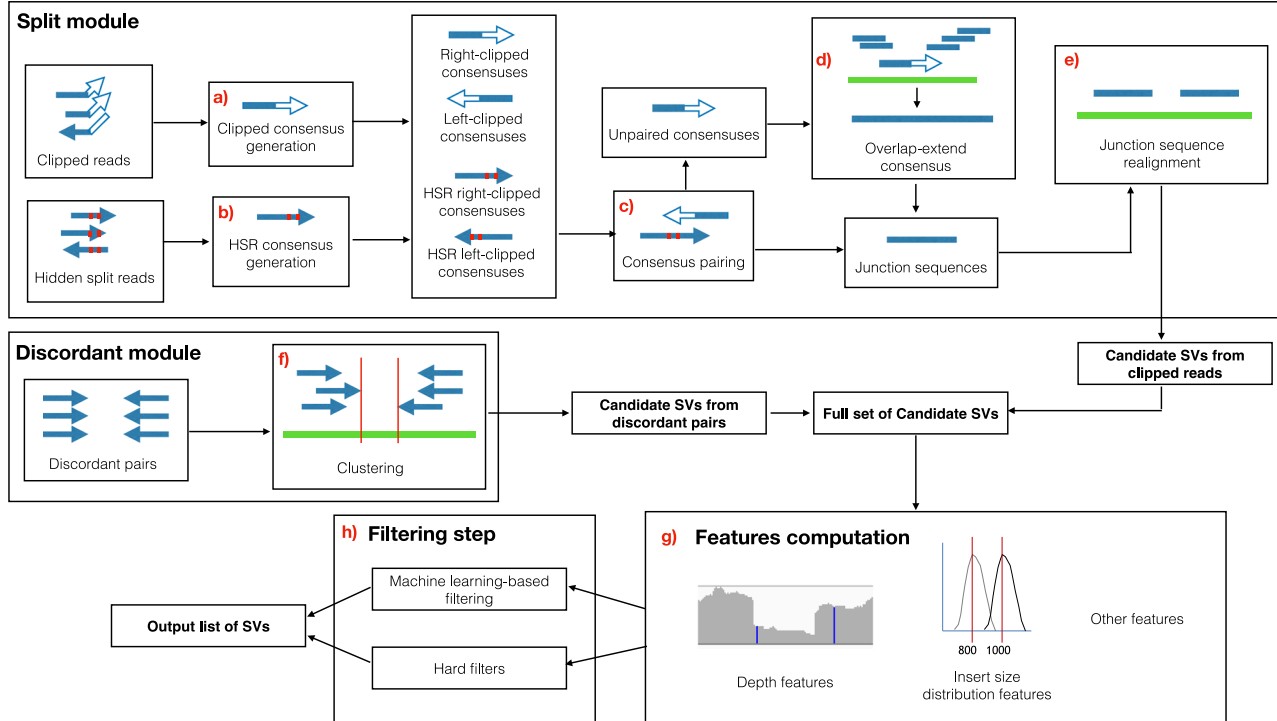

**Fig. 3 | Overview of the SurVIndel2 method.** It primarily identifies candidate copy number variants (CNV) by using split reads and hidden split reads. Step (**a**) piles up split reads into consensus sequences, and step (**b**) does the same with hidden split reads. Consensus sequences that potentially represent the same CNV are paired in step (**c**) to generate a junction sequence, which is a subsequence of the alternative allele containing a breakpoint of the CNV. Consensus sequences that could not be paired are extended into a junction sequence using the reads mapped close to or overlapping the reads forming the consensus sequence in step (**d**). Junction sequences are then realigned onto the reference genome to discover the actual CNV in step (**e**). Candidate CNVs that do not generate clipped reads are identified as clustering discordant pairs in step (**f**). In step (**g**), a series of features are calculated for each candidate CNV, mainly by (i) statistically testing the distribution of the read pairs surrounding the CNV and (ii) testing changes in read depth in the CNV region as well as its flanking regions. Finally, these features are used in step (**h**) to exclude the CNVs that are likely false positives by using a machine learning model or, if unavailable, hard filters.

calculate a putative range for the location of the second breakpoint (Supplementary Fig. 1): we call this range *opposite breakpoint range*. We pair the consensuses so that each pair contains a left-clipped and a right-clipped consensus, and the consensuses are within each other's opposite breakpoint range (Fig. 3c). For each pair we generate a *junction sequence* by overlapping the left-clipped and right-clipped consensuses. The junction sequence is expected to be the DNA sequence at the precise junction where two different genomic regions have been joined together as a result of the variant (the left and right-flanking regions for deletions, the two copies of the duplicated region for tandem duplications, examples in Fig. 1a, b). For consensuses that are not paired, we attempt to generate the junction sequence by extending them using an overlap-layout-consensus approach (Fig. 3d). Finally, we realign the junction sequence to the reference genome to obtain the precise breakpoints of the CNV (Fig. 3e).

The discordant module is considerably simpler, and it clusters discordant pairs in order to detect CNVs (Fig. 3f). One drawback is that, unlike split reads, discordant pairs often provide an approximate position for the breakpoints, rather than a precise one. When the same CNV is predicted by both the split module and the discordant module, the coordinates from the split module are retained, since they are deemed precise.

## HGSVC2 benchmark
As mentioned in Section 2.1, the Human Genome Structural Variation Consortium (HGSVC) has published a catalogue (called HGSVC2) of insertions and deletions in 35 human genomes (34 from the 1000 genomes project, plus HG002), identified from long-read PacBio whole-genome sequencing and Strand-seq data[11]. Furthermore, the New York Genome Centre (NYGC) has recently performed high-coverage, PCR-free sequencing for the 2504 original samples in the 1000 Genome project, along with 698 additional samples. We downloaded the CRAM files published by NYGC for the 34 samples in the HGSVC2 benchmark (30x coverage, except for one sample, which had 60x coverage). We also obtained a 50x short-read dataset for HG002 by downloading runs from SRR1766442 to SRR1766486 and mapping them to hg38 using BWA-MEM. We use these datasets to benchmark the performance of SurVIndel2, and three highly cited methods that performed well in a recent benchmarking analysis[7]: Delly[12], Lumpy[13] (which we ran using the Smoove package, as recommended by the authors) and Manta[14]. These SV callers were used in some of the major studies on SVs in human populations[2,4,5]. We also included the original SurVIndel[3] in the comparison. See Methods for more details on the selection of the callers and the versions used.

In the HGSVC2 benchmark, tandem duplications are simply reported as insertions. Since we are interested in calculating the sensitivity only on tandem duplications, we used the method introduced in[15] to identify them from other insertions. See Methods for details on how sensitivity and precision are calculated.

The machine learning model of SurVIndel2 was trained using the 34 samples in HGSVC2 sampled by the NYGC. For this reason, when testing them, we use a leave-one-out strategy. More precisely, for each sample S in HGSVC2, we train a separate model by excluding S from the training set, and we use it to filter the raw calls by SurVIndel2 detected from S.

SurVIndel2 outperforms the other methods in both deletions and duplications, often by a large margin (Fig. 4a, b). For deletions, Manta is the only method that almost matches SurVindel2 precision (average

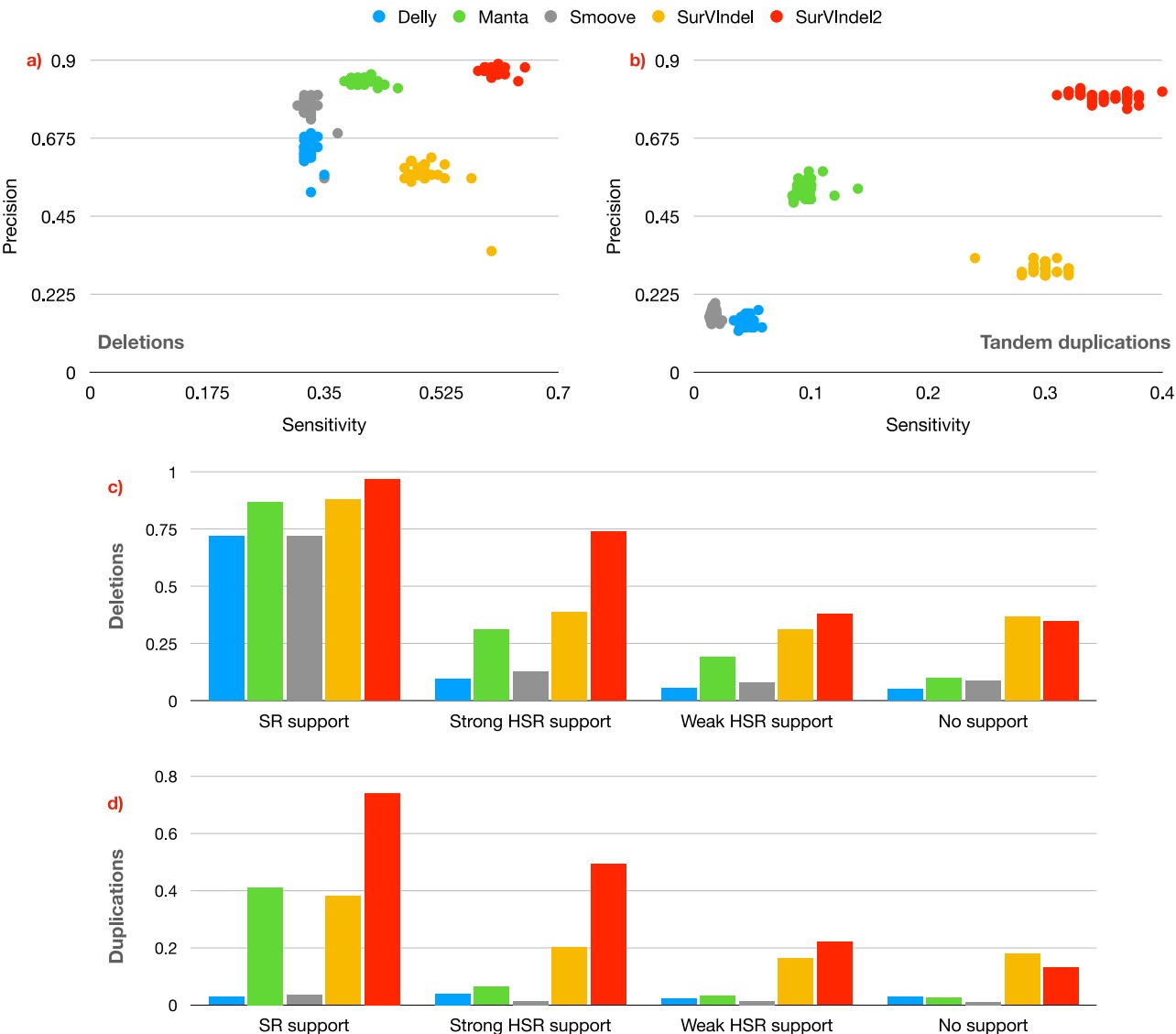

**Fig. 4 | Performance of the tested methods on the HGSVC2 benchmark.** Sensitivity and precision for deletions (**a**) and tandem duplications (**b**) in the HGSVC2 catalogue. Each sample is a dot, its *x* coordinate is the sensitivity while its *y* coordinate is the precision. SurVIndel2 has higher sensitivity and precision than the other methods. **c** Sensitivity of the methods in predicting deletions in HG002, stratified by supporting evidence. Split reads (SR)-supported deletions are relatively easy for all methods, although SurVIndel2 has a clear advantage. However, SurVIndel2 is the only method that can detect most deletions with strong hidden split reads (HSR) support. **d** For tandem duplications, existing methods perform poorly even for SR-supported events. SurVIndel2 can detect nearly twice as many duplications as existing methods that are supported by split and hidden split reads.

precision is 0.84 for Manta, 0.87 for SurVIndel2), but at the expense of greatly reduced sensitivity (average sensitivity is 0.41 for Manta, 0.60 for SurVIndel2, 46% increase). SurVIndel is the second most sensitive method (0.5 average sensitivity and average precision 0.57), but it is considerably less precise than both Manta and SurVIndel2.

Tandem duplications are far more difficult to predict than deletions using short reads. SurVIndel2 is the most sensitive and precise method (average sensitivity 0.35 and average precision 0.8). The next best method in terms of sensitivity is SurVIndel (average sensitivity 0.3 and average precision 0.3). The next best method in terms of precision is Manta (average sensitivity 0.10 and average precision 0.53).

The datasets used for training the model are very homogeneous: they were all sequenced by one sequencing centre (NYGC), with the same platform (Illumina NovaSeq 6000), with the same sequencing depth (30×, except for one which was 60×), which can lead to overfitting. However, the HG002 sample confirms that this is not the case for our model. HG002 was sequenced by a different entity (GIAB

Consortium) using a different platform (Illumina HiSeq 2500), and its depth is different (50×). SurVIndel2 performs just as well on it: sensitivity and precision were 0.64 and 0.84 for deletions, 0.37 and 0.76 for duplications. We also tested the performance of the methods on two additional benchmarks for the HG002, both based on the hg19 reference. The first benchmark, which we call GIAB v0.6 benchmark, was generated by Zook et al.[16]. The second benchmark, which we call the Sniffles2 benchmark, was obtained by running Sniffles2[17] on HiFi reads for HG002, and retaining deletions and duplications of size at least 50 bp (Supplementary Fig. 7). The performance of the callers are in line with what was observed on the HGSVC2 benchmark, with SurVIndel2 performing the best overall.

Furthermore, we compared SurVIndel2 to the SV calling module in DRAGEN, a proprietary caller based on Manta that, unlike the other tools evaluated, is not open source and requires a license for use. Despite these limitations, DRAGEN is considered a state-of-the-art tool due to its prominence in the field and high accuracy. SV callings for the

samples in HGSVC2 were published by Behera et al.[18]. Our benchmarks demonstrate that although DRAGEN is a marked improvement over Manta, it does not achieve the same level of sensitivity and precision as SurVIndel2 (Supplementary Fig. 8). In particular, sensitivity for deletions is improved from 0.41 average of Manta to 0.47 for DRAGEN (0.60 for SurVIndel2), while precision remained the same (0.84 average for both Manta and DRAGEN, 0.87 for SurVIndel2). The improvement on tandem duplications was even more marked, involving both sensitivity (0.10 average for Manta, 0.145 for DRAGEN) and precision (0.53 average for Manta, 0.68 for DRAGEN); however, SurVIndel2 was still more than twice as sensitive and also more precise (0.35 average sensitivity and 0.8 average precision).

In Section 2.1 we have partitioned the CNVs in HG002 depending on whether they were supported by split reads, hidden split reads, or unsupported. In order to maintain a high precision, our algorithm only considers HSRs that have at least three differences with the reference genome (an indel or a mismatch relative to the reference is counted as one difference, see "Methods"). Let us call such HSRs as *strong HSRs*. Therefore, we expect that SurVIndel2 will be able to detect HSR-supported CNVs that have good support from strong HSRs, but not the rest. To study this, we partition HSR-supported CNVs into two categories: CNVs with strong HSR support (supported by at least 5 strong HSRs) and with weak HSR support (supported by less than 5 strong HSRs). 53% and 67% of the HSR-supported deletions and duplications, respectively, have strong HSR support.

All tested methods tend to perform well in detecting deletions that are supported by split reads (Fig. 4c). SurVIndel2 detects nearly all true positive deletions in this category (0.97 sensitivity). As expected, SurVIndel2 has a large advantage over the other methods in finding deletions supported by strong HSRs (0.74 sensitivity for SurVIndel2, 0.39 for SurVIndel). Predictably, detection power drops for deletions with weak or no support, although SurVIndel2 and the original Sur-VIndel are still the most sensitive methods. In particular, focusing on SurVIndel2 calls that predict unsupported benchmark deletions, we observed that 38% of them are supported detecting using HSRs. We identified three possible reasons why such deletions may have been classified as unsupported: (1) the alternative allele assembled by spoa might not have been 100% correct ("Methods"), or (2) the number of HSRs is less than 5 (SurVIndel2 is able to detect CNVs supported by as few as 3 reads, as long as either all three are right-clipped or all three are left-clipped); or (3) when using BWA to map short reads to the alternative alleles, the reads might have been mapped somewhere else or unmapped. Another 38% are predicted by the discordant pairs module, therefore no SR or HSR support is required. Out of the remaining 24% that are supported by split reads, nearly half are supported by less than 5 split reads, and are hence classified as unsupported.

Surprisingly, SurVIndel2 has a large advantage not only for tandem duplications supported by strong HSRs but also for those supported by split reads (Fig. 4d). Methods other than SurVIndel and SurVIndel2 are essentially unable to call duplications not supported by split reads, which explains their poor sensitivity.

We observed from Fig. 2c and d that CNVs outside of repetitive regions are more likely to be supported by split reads, while the CNVs within repetitive regions are mostly supported by hidden split reads or are unsupported. The proportion of split reads support, hidden split reads support, and no support in repetitive regions is similar between deletions and duplications; however, because a much larger proportion of duplications is within repetitive regions (Fig. 2a: 64% of deletions and 93% of duplications are in tandem duplication regions), it follows that a larger portion of duplications is not supported by split reads compared to deletions. This might explain, together with our observations from Section 2.7, why tandem duplications are so difficult to call compared to deletions.

## Performance on different organisms

The filtering model we provided was trained on the CNVs called from the 34 human samples. We call this the *default model*. For this reason, we are interested in (i) how SurVIndel2 performs on different organisms using the default model and (ii) what are the implications on the sensitivity and precision of the method if a more targeted training set is used (i.e., the model is retrained using data from the target organism). To address these questions, we conducted an analysis by running the callers on seven *Arabidopsis thaliana* samples that were sequenced as part of the 1001 Genomes Project. These samples were mapped to the TAIR10 reference genome. We generated a benchmark dataset by using Sniffles2 on PacBio HiFi reads for the same samples. As we did for HGSVC2, we used the method in[15] to separate tandem duplications from other insertions. For SurVIndel2, we used two classification models: one trained on the 34 human samples with long reads in the 1000 Genome Project (Section 2.3), and one trained on the seven *Arabidopsis thaliana* using a leave-one-out cross-validation (i.e., when classifying sample S, sample S is excluded from the training set).

SurVIndel2 outperforms all of the existing methods (Fig. 5a). Two key points that must be noted are (a) when trained on *Arabidopsis thaliana* data, SurVIndel2 shows noticeably improved performance in calling *Arabidopsis thaliana* CNVs compared to when trained on human data and (b) SurVIndel2 trained on human data still outperforms existing methods. This means that if a reliable catalogue of CNVs (i.e., obtained from long reads) is available for the target organism, it is possible to increase the performance of SurVIndel2; however, if such a catalogue is not available, the default model works well.

We also selected samples of other species (*Bos taurus*[19], *Mus musculus*[20] and *Oryza sativa*[21,22]) for which both Illumina paired reads and PacBio HiFi reads were available. All of the selected organisms are important either as food sources or as model organisms. Once again, SurVIndel2 has the best performance for every sample, both for deletions and duplications (Fig. 5b). We expect that the performance would be even better if the model was trained for the specific organism, as in the case of *Arabidopsis thaliana*.

## A catalogue of deletions and tandem duplications for the 3,202 genomes in the 1000 genomes project

We used SurVIndel2 to generate a catalogue of deletions and tandem duplications for 3202 samples from the 1000 g genomes project sequenced by the NYGC. SurVIndel2 was run on each individual sample, and the calls were left-aligned and clustered across samples ("Methods"). The final catalogue contained 247,822 deletions and 232,227 tandem duplications. 205,258 deletions (82.8%) and 189,413 duplications (81.6%) were rare (detected in 1% of the population or less), and 90,137 deletions (36.4%) and 74,369 duplications (32.0%) were singletons (detected in only one individual). Africans had more deletions (Fig. 6a) and duplications (Fig. 6b) compared to non-Africans: the median count was 7509 deletions and 5129 duplications for Africans vs 6294 deletions and 4478 duplications for non-Africans. The number of events was similar in the remaining superpopulations. Fig. 6c shows the size distribution for the deletions and duplications predicted. The peaks associated with ALU (~ 310 bp) and LINE (~ 6 kbp) deletions are clearly visible. PCA is able to separate the superpopulations (Fig. 6d). This is also true when performing the PCA using deletions and tandem duplications separately (Supplementary Fig. 18). Furthermore, PCA was able to segregate the superpopulations into subpopulations (Supplementary Fig. 19).

The NYGC also generated a catalogue of SVs (including deletions and tandem duplications)[5] by integrating multiple callers, including Manta. We refer to this catalogue as 1000 g-SV. Fig. 7 shows a comparison between 1000 g-SV and our catalogue. Similarly to Section 2.3, we measured sensitivity and precision for 1000 g-SV and for SurVIndel2 on the 34 samples in HGSVC2. SurVIndel2 had consistently higher sensitivity and precision. For deletions (Fig. 7a), SurVIndel2 was, on

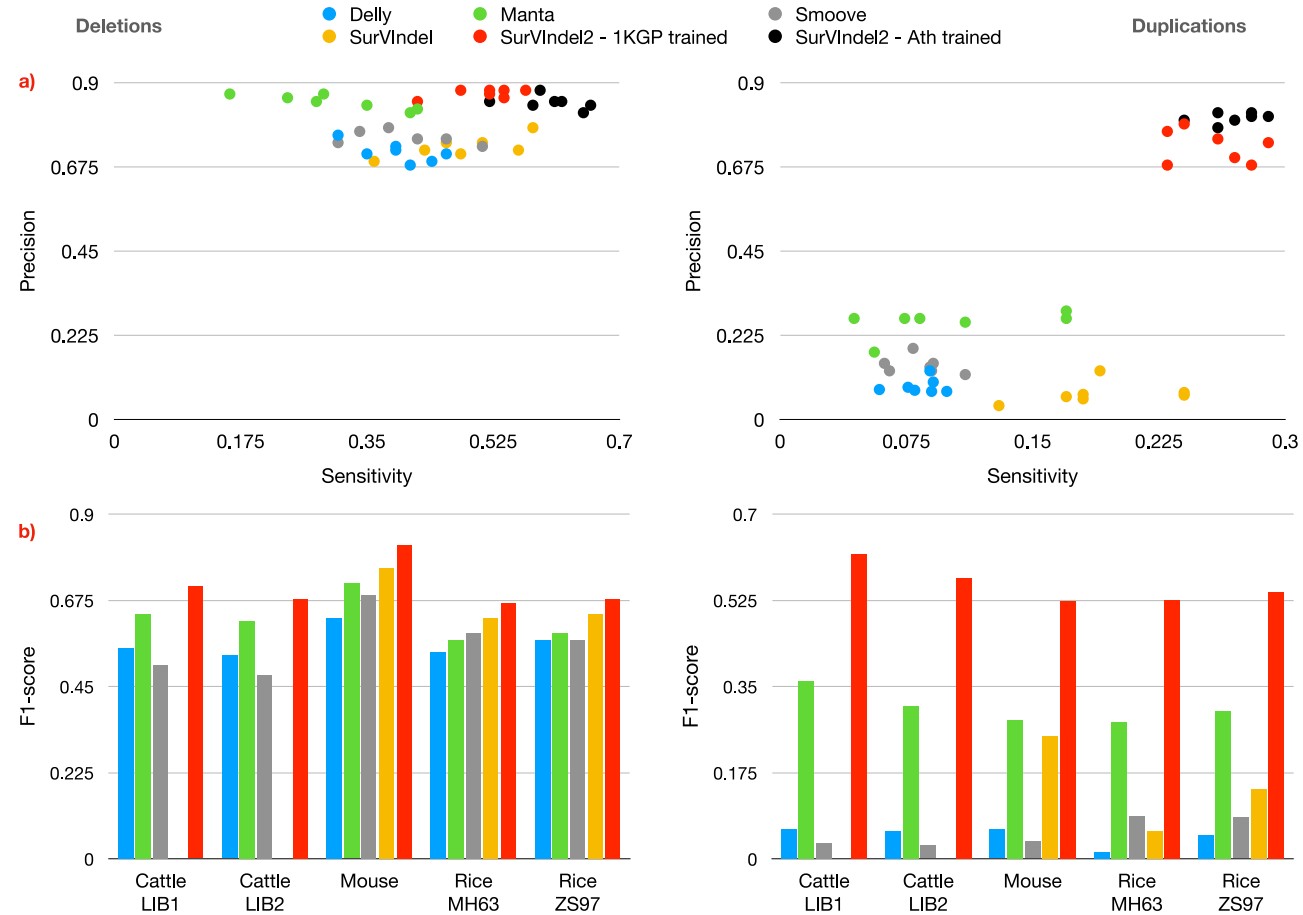

**Fig. 5 | Performance of the tested methods on non-human organisms.**
**a** Sensitivity and precision for deletions (left) and tandem duplications (right) in the *Arabidopsis thaliana* benchmark. Unsurprisingly, the *Arabidopsis thaliana*-trained SurVIndel2 performs the best; however, the human-trained SurVIndel2 still performs better than the existing methods. **b** F1-score for deletions (left) and duplications (right) for five non-human libraries: two *Bos taurus* (Cattle LIB1 and Cattle

LIB2), one *Mus musculus* (Mouse) and two *Oryza sativa* (Rice Minghui 63 and Rice Zhenshan 97). The legend of the panel (**a**) applies. Benchmark catalogues were obtained by running Sniffles2 on PacBio HiFi reads. SurVIndel2 performs better for every organism. The advantage is particularly evident for tandem duplications. SurVIndel failed to run for the two Cattle libraries.

average nearly twice as sensitive (average sensitivity was 0.345 for 1000 g-SV and 0.60 for SurVIndel2), while being more precise (average precision was 0.78 for 1000 g-SV and 0.87 for SurVIndel2). For tandem duplications (Fig. 7b), SurVIndel2 was more than 4 × more sensitive (average sensitivity was 0.084 for 1000 g-SV and 0.35 for SurVIndel2) and significantly more precise (0.55 for 1000 g-SV, 0.80 for SurVIndel2). When comparing the full catalogues (Fig. 7c, d), SurVIndel2 captures the vast majority of the deletions in 1000 g-SV, and deletions in 1000 g-SV but not predicted by SurVIndel2 have low validation rate according to HGSVC2 (0.14). Furthermore, SurVIndel2 predicts 130,657 deletions not present in 1000 g-SV, with a high validation rate (0.87). The difference in detected tandem duplications is even more striking. Duplications predicted by 1000 g-SV but not by SurVIndel2 have again very low validation rate (0.19). The two datasets share 35,219 duplications, and SurVIndel2 predicts an additional 197,008 duplications, with a high validation rate (0.85). Overall, it appears that SurVIndel2 is able to predict a more accurate and significantly more complete catalogue of both deletions and tandem duplications.

**SurVIndel2 enhances small indels callers**
Although SurVIndel2 was developed and tuned to detect SVs (≥ 50 bp), its approach is theoretically able to recover some of the smaller indels, as long as they produce a sufficient number of split or hidden split reads. Therefore, we investigated whether SurVIndel2 is able to

complement and complete the calls provided by DeepVariant since it has been found the best-performing tool for small indels[23,24]. We focused on indels of size between 30 and 50 bp since smaller indels are unlikely to generate enough evidence for SurVIndel2 to use.

In order to measure the sensitivity of the methods, we use the benchmark catalogue for small indels provided by HGSVC2. However, we noticed that this benchmark underestimates the precision of the methods. Upon manual inspection using PacBio HiFi reads and IGV[25], most of the calls by DeepVariant or SurVIndel2 that were missing from the benchmark were clearly supported by the long reads. For this reason, in order to more accurately measure precision, we created a complete benchmark for both HG00512 and HG002 by concatenating three callsets: (i) HGSVC2,(ii) Sniffles2 calls on PacBio HiFi reads, and (iii) DeepVariant calls on PacBio HiFi reads.

Figure 8a and Supplementary Fig. 20 illustrate the improved sensitivity achieved by using both DeepVariant and SurVIndel2 for the detection of deletions. On average, the sensitivity for DeepVariant alone was 0.6, whereas this increased to 0.71 when SurVIndel2 was added. This increase was not at the expense of precision, which stayed the same (0.97).

The improvement in sensitivity was even more pronounced for small insertions (Fig. 8b): by adding SurVIndel2 the average sensitivity increased from 0.38 to 0.52. There was an apparent decrease in precision, from 0.99 with DeepVariant alone to 0.93 when combined with

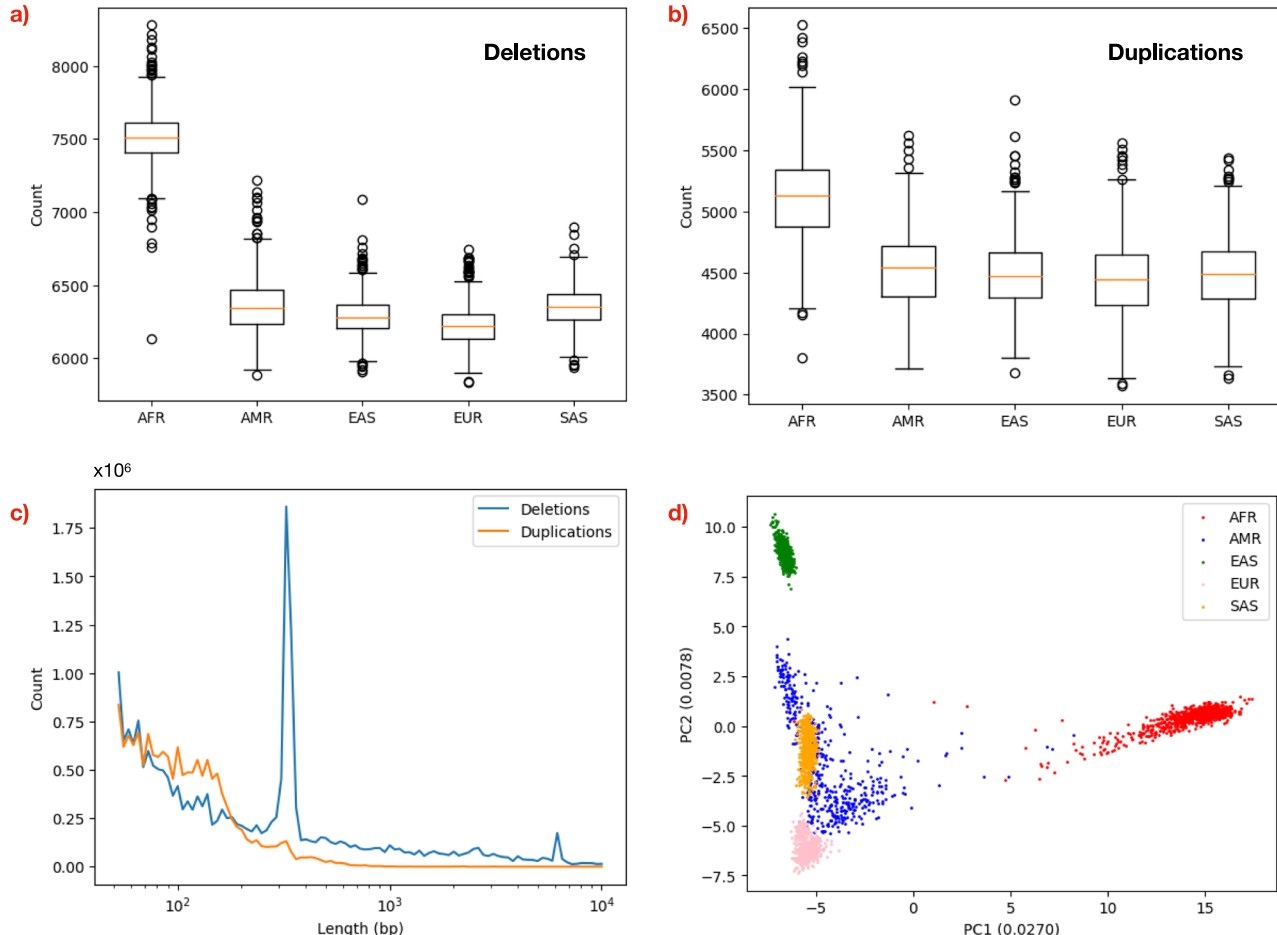

**Fig. 6 | Statistics for the catalogue of CNVs called by SurVIndel2 on the 1000 Genomes Project. a, b** Count of deletions and duplications in different super-populations. Africans have more deletions and duplications compared to non-Africans: the median count was 7509 deletions and 5129 duplications for Africans vs 6294 deletions and 4478 duplications for non-Africans. The boxes contain values from the lower to the upper quartile, the line within the box is the median, and the whiskers extend by 1.5 times the interquartile range. Circles represent data points outside of the whiskers. **c** The size distribution for deletions and duplications detected in 1000 g, adjusted by their frequency in the population. For deletions, the expected peaks associated with ALU (~310bp) and LINE (~6000bp) deletions are visible. **d** Principal component analysis is able to clearly separate the superpopulations.

SurVIndel2 for HG00512, and from 0.97 to 0.92 for HG002. However, upon manual inspection of PacBio HiFi reads using IGV, we noticed that a significant portion of the insertions marked as false positives had clear support from the alignment of the reads. In particular, we randomly selected 30 examples of insertions marked as false positives in HG00512, and for 15 of them (50%), the HiFi long reads supported the presence of an insertion (IGV views available for all 30 in Data Availability). Similarly, for HG002, 17 out of 30 (57%) of the apparent false positives had clear support from the HiFi long reads (IGV views available for all 30 in Data Availability). Consequently, the projected precision for the DeepVariant and SurVIndel2 combination is recalibrated to be ~0.975 for HG00512 and 0.966 for HG002, which suggests that the decrease in precision was far more modest than it initially appeared.

Overall, our findings indicate that the integration of SurVIndel2 with DeepVariant can provide a remarkably sensitive and precise catalogue of variants for an individual, from very small to very large, far more complete than what was possible without SurVIndel2.

**Reasons for unsupported CNVs**

Many existing CNVs have no support. In particular, in Fig. 2c, d, we showed that 41% of deletions and 36% of duplications in tandem repetitive regions in HG002 have no sufficient support from split and hidden split reads. We will refer to them as *unsupported CNVs*. This section aims to find the reasons behind the lack of supporting reads for unsupported CNVs.

As mentioned in Section 2.1, we expect that the major cause of unsupported CNV is that the sequences flanking the junction sequence created by the CNV (deletion or duplication) is not a novel sequence. Every read sequenced from such junction sequence will perfectly align to the reference genome, hence it will not support the existence of a CNV. This scenario is illustrated in Fig. 1c, d.

To formalise this notion, we present a probability measure for each CNV, which we call the *expected support (ES) score*. The ES score for a CNV represents the likelihood that a read which contains a breakpoint of the CNV will provide evidence for the presence of the CNV when it is aligned to the reference sequence (i.e., it will be a split or a hidden split read). When ES is 0, we do not expect any read to support the CNV, while an ES of 1 means that we expect every read containing a breakpoint to support the CNV. The calculation of the ES value is described in "Methods". We expect that most unsupported CNVs will be due to a very low ES score.

A large portion of the unsupported deletions in HG002 do indeed have low ES scores (Fig. 9a, left), while supported deletions tend to have much higher ES values (Fig. 9a, right). However, more than 40% of the unsupported deletions have high ES (≥ 0.5), which suggests that a

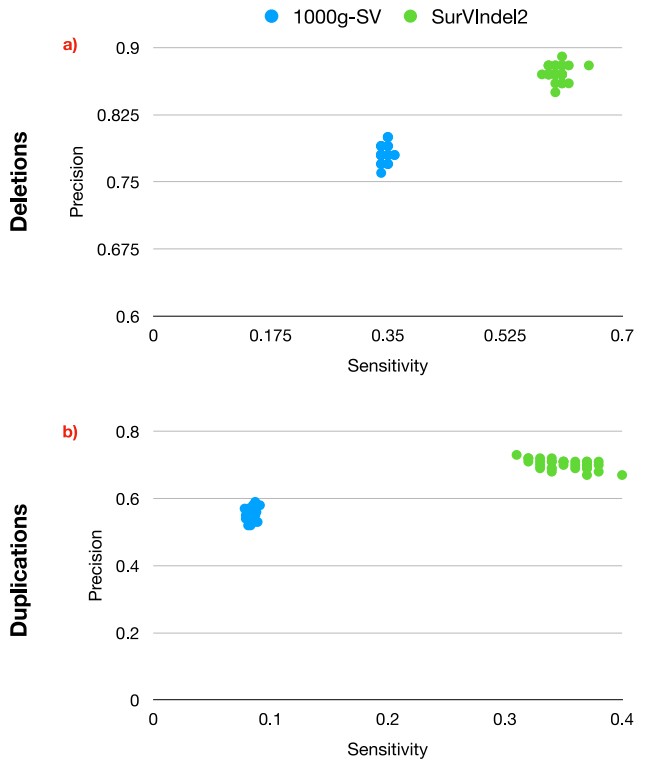

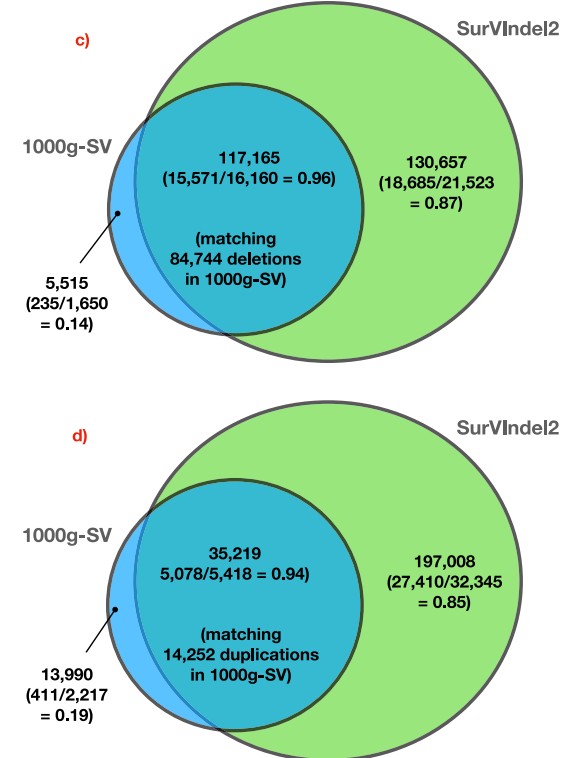

**Fig. 7 | Comparison between the 1000 g-SV and SurVIndel2 datasets.**
**a**, **b** Sensitivity and the precision for the 34 samples in HGSVC2 for deletions (**a**) and tandem duplications (**b**). The dataset produced by SurVIndel2 is both more sensitive and more precise than the 1000 g-SV dataset. **c** Overlap between deletions in 1000 g-SV and SurVIndel2 datasets. Between parentheses, the validation rates of calls in samples with long reads. Not only does SurVIndel2 cover nearly all of the deletions in 1000 g-SV, but it predicts 130,657 deletions that are not present in the latter. The validation rate for SurVIndel2-unique events is far higher than the validation rate of 1000g-SV unique events. **d** Similarly, SurVIndel2 predicts a large portion of the tandem duplications in 1000 g-SV, and the ones that are not predicted have a very low validation rate. SurVIndel2 also predicts 197,008 duplications that are not present in 1000 g-SV, with a high validation rate.

low ES is not the only reason for lack of support. We found that many deletions are unsupported because the reads sequenced from the breakpoint region are very low in quality. Supplementary Figs. 9 and 10 show examples of such deletions.

To demonstrate how common this issue is, we built high-quality alternative alleles for a subset of CNVs in HG002 ("Methods"). Then, we used BWA MEM to map the paired reads of HG002 (Illumina HiSeq 2500) to the alternative alleles, and, for each CNV, we counted the reads correctly sequenced (i.e., without mismatches or indel) that contain a breakpoint. Given that the dataset has a coverage of about 50x, we expect the number of reads covering the breakpoint of the deletion to be around 25 for a heterozygous deletion and 50 for a homozygous one. Fig. 9b shows that unsupported deletions with high ES tend to have fewer reads covering the breakpoint than expected: 91% of them have < 10 reads, and 64% have < 5. Conversely, supported deletions tend to have more reads containing the breakpoints (average 18.4 for supported high ES deletions, 3.7 for unsupported high ES deletions). Therefore, two ingredients determine the level of support for a deletion: the ES score and the number of reads that contain the breakpoint.

Surprisingly, unlike unsupported deletions, we observed that most unsupported tandem duplications (77%) have a high ES score (Fig. 10a). Only 16% have a low ES score. In fact, excluding the very low ES values (0-0.1), the distribution of ES score is similar between unsupported (left) and supported (right) duplications. The issue of poorly sequenced breakpoint regions is even more prevalent in duplications than it is in deletions: 96% have < 10 reads, and 85% have < 5. As for deletions, supported duplications tend to have a much higher number of reads containing a breakpoint (average 16.6 for

supported high ES duplications, 1.6 for unsupported high ES duplications).

Repeating these experiments on the HG00512 sample sequenced by the NYGC using a different platform (Illumina NovaSeq 6000) yields the same conclusions (Supplementary Figs. 11, 12). These experiments demonstrate that there is a serious issue in sequencing a large number of the regions containing CNVs, and variant calling is unlikely to further improve significantly unless such issues are mitigated.

To reinforce our hypothesis, we create a synthetic dataset for HG002, which we refer to as HG002-SIM, that closely resembles actual sequencing data but does not have the issue of poorly sequenced regions. We achieve this by using the ART read simulator[26] to simulate an Illumina HiSeq 2500 50x dataset from a high-quality HG002 T2T assembly[27,28]. The results match our expectations: nearly all unsupported deletions and duplications have low ES (Supplementary Fig.s 13, 14), and their number is greatly reduced. For this reason, the sensitivity of SurVIndel2 is noticeably higher on the synthetic dataset, especially for duplications (Supplementary Figs. 15, 16).

In conclusion, our findings reveal that lack of evidence for the existing CNVs is only partially due to the regions containing the CNVs being very repetitive. Another major reason is that the sequencing of many such regions is extremely noisy. This seems to be especially true for regions prone to tandem duplications. We have shown through a realistic synthetic dataset that without this issue, the number of CNVs detectable using short reads would be much higher.

## Discussion
Existing CNV callers rely on read depth, split reads and discordant pairs when detecting CNVs from short reads. In this paper, we described

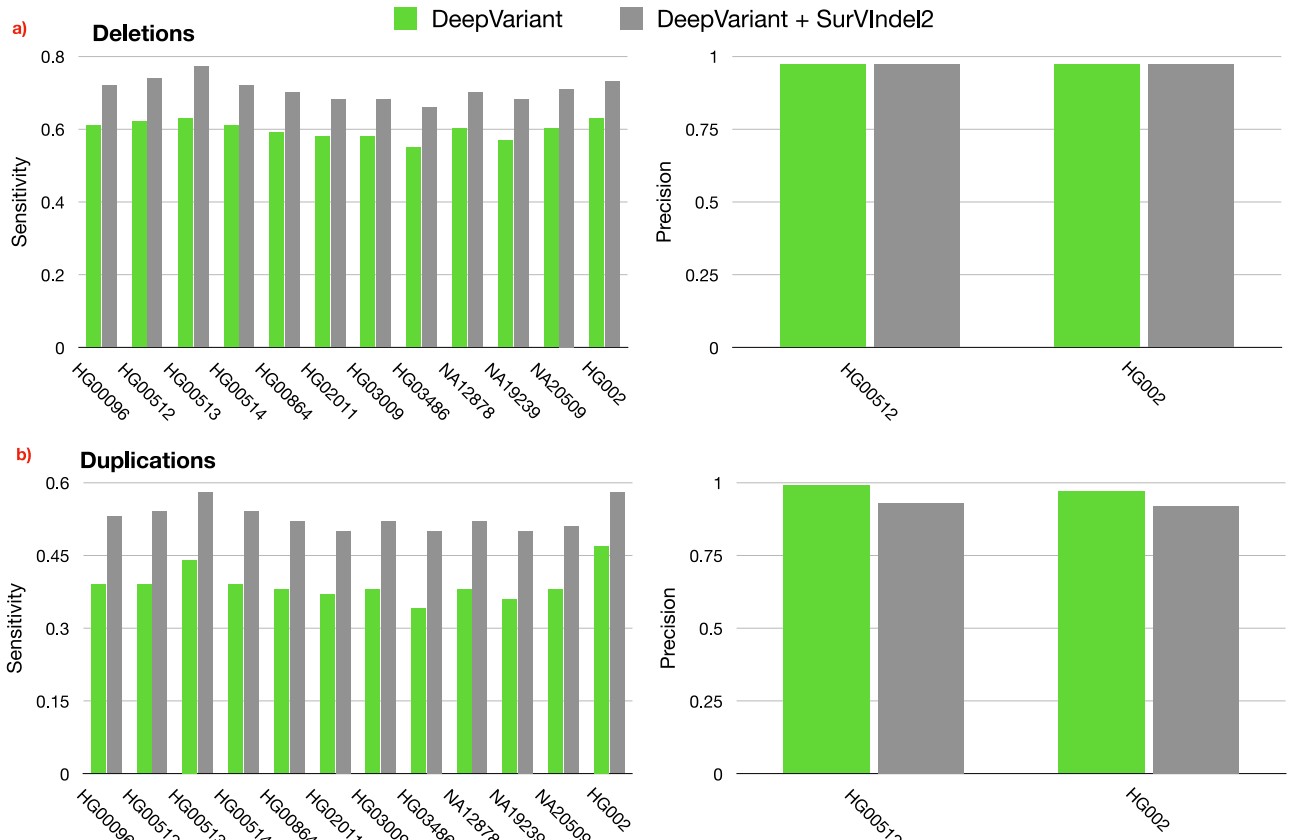

**Fig. 8 | SurVIndel2 recovers small indels (30 to 50 bp) missed by DeepVariant.**
**a** By using both DeepVariant and SurVIndel2, we are able to recover more true positive small deletions compared to using DeepVariant alone (average sensitivity was 0.6 for DeepVariant, 0.71 for DeepVariant plus SurVIndel2). The precision was not changed (0.97 for both). **b** The increase in sensitivity was even greater for small insertions. The average sensitivity for DeepVariant was 0.38, which increased to 0.52 when introducing SurVIndel2. Precision apparently decreased from 0.99 to 0.93; however, when manually examining 30 random calls marked as false positives, 15 of them showed clear support from PacBio HiFi reads. Therefore, the projected precision of DeepVariant plus SurVIndel2 is 0.975. Supplementary Figs. 20, 21 show the sensitivity for all 35 samples in HGSVC2.

how CNVs in repetitive regions often do not generate such evidence. For this reason, we introduced a new type of evidence, called hidden split reads. By using it, we are potentially able to discover more than double the number of deletions and duplications compared to only using split reads. From this observation, we developed SurVIndel2, a CNV caller that uses split reads and discordant pairs, as well as hidden split reads, to produce a comprehensive set of candidate CNVs. SurVIndel2 borrows the idea of statistical testing from its predecessor, SurVIndel, and uses it in a machine-learning filtering algorithm to produce a comprehensive and accurate catalogue of CNVs.

We used publicly available benchmarks to show that SurVIndel2 outperforms the state of the art in CNV calling. On the HGSVC2 benchmark, a comprehensive catalogue of the CNVs in 35 human samples, SurVIndel2 was the most sensitive and precise caller for both deletions and duplications, often by a large margin. The improvements were particularly large for duplications since all of the existing methods had either very low sensitivity or very low precision.

Our machine learning filtering was trained on human data, so we were interested in two key questions: how does it perform on non-human data, and is it possible to improve the performance by training for the target organism. We investigated this using a variety of different organisms, and we learned that (i) the human model works well for every tested organism and (ii) the performance increases even further when the model is trained for the specific organism.

To demonstrate how SurVIndel2 is useful in practice, we used it to generate a catalogue of deletions and tandem duplications for the 1000 Genome Project and compared it to a recently published catalogue generated using state-of-the-art pipelines. Our catalogue revealed hundreds of thousands of events that were not present in the existing one, and the long read-based validation rates for those events were high. We expect that SurVIndel2 will be beneficial to the community and will help uncover a large number of CNVs that would be missed by using existing tools while maintaining the number of false positives to a minimum.

We also examined the ability of SurVIndel2 to complement small variant callers, in particular DeepVariant. Our findings indicate that incorporating SurVIndel2 with DeepVariant significantly improved the sensitivity for both deletions and small insertions detection. Notably, this increase in sensitivity did not compromise precision. Using DeepVariant with SurVIndel2 provides a view of the full spectrum of variants in an individual, from small to large, that is more precise and complete than what was previously possible.

Finally, we investigated why a large portion of CNVs lacked support from both split and hidden split reads. We found that the reads in the breakpoint regions of many CNVs were extremely noisy, and for many CNVs, not a single correctly sequenced read contained a breakpoint. This was especially serious for tandem duplications, and we have also shown by simulation that if this issue was not present, the number of true positive duplications predicted would greatly increase. Although it is known that some genomic regions are difficult to sequence, this was the first study, to our knowledge, that systematically investigated the magnitude of the problem and its impact on CNV calling.

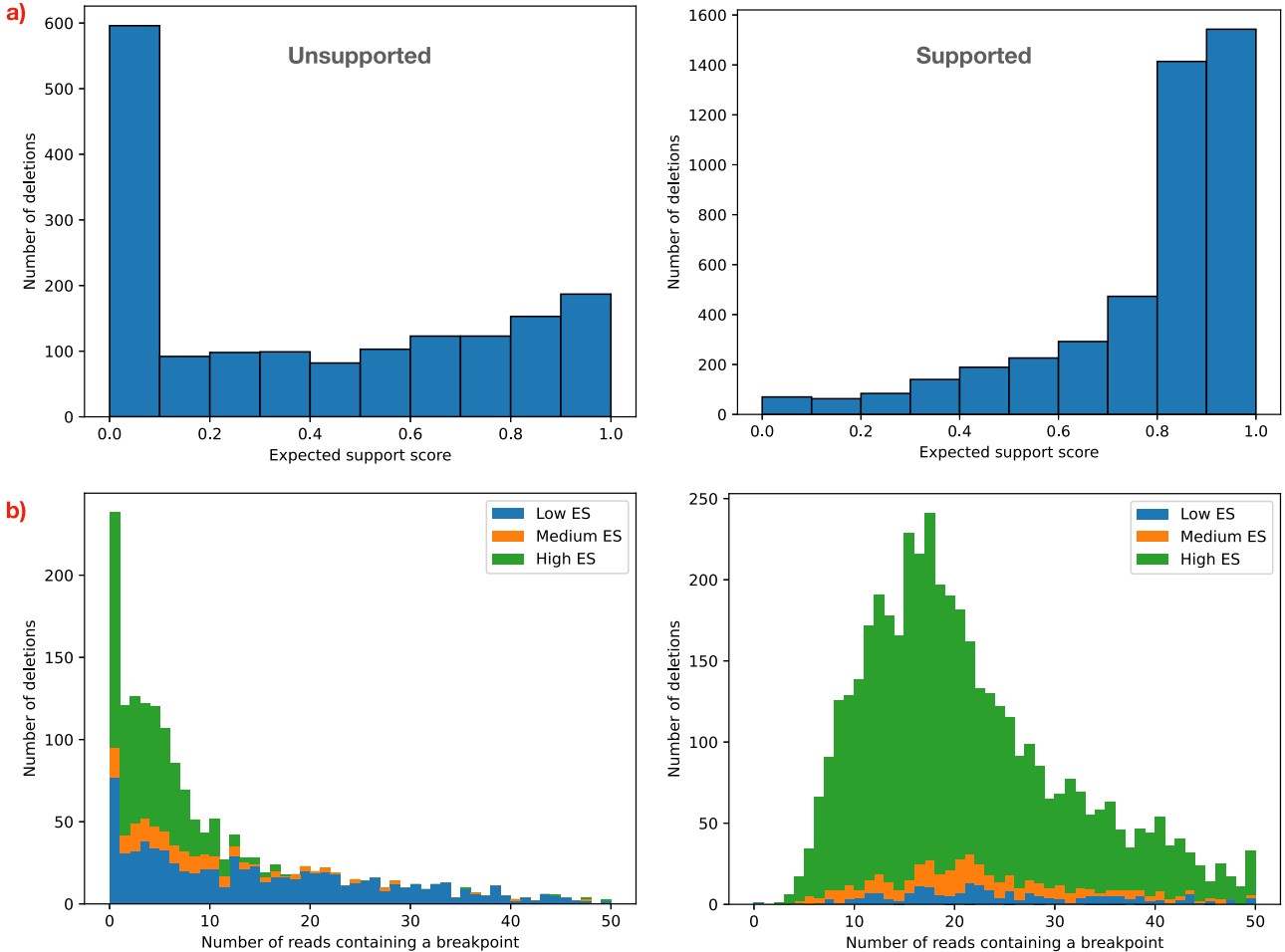

**Fig. 9 | Factors that contribute to absence of support for deletions in HG002.**
**a** As expected, a large portion of unsupported deletions have a low expected support (ES) score (left). ES score is the probability that a read containing the breakpoint of the deletion will support its existence, i.e., it will be either a split or a hidden split read. On the other hand, most deletions with support have very high ES scores (right). However, a low ES score is not the only factor that contributes to the lack of support: more than 40% of the unsupported deletions have an ES score ≥ 0.5. **b** Another factor is a lack of correctly sequenced reads containing the breakpoint of the deletions. For each deletion with a high-quality alternative allele

("Methods"), we mapped the short reads of HG002 to the alternative allele and counted the reads correctly sequenced (i.e., without mismatches or indels) and containing the deletion breakpoint. We further stratified the deletions into low ES (< 0.3), medium ES (≥ 0.3 and < 0.5) and high ES (≥ 0.5). High ES unsupported deletions (left) tend to have very few, or no reads correctly sequenced that contain the breakpoint, while deletions with a higher number of reads containing the breakpoint have low ES. Supported deletions (right), on the other hand, have both ingredients necessary for support: reads containing the breakpoint, and high ES.

Many variant callers, including some of those examined in this study, are able to jointly examine a tumour and a normal sample and identify somatic variants. This is an important functionality, especially in cancer research, which SurVIndel2 does not currently support. Somatic variants could be identified by determining variants independently on the tumour and the normal samples and then searching for those only appearing in the tumour. However, we believe that examining tumours and normal samples jointly has the potential to produce better results. Extending the SurVIndel2 algorithm to effectively do so will be an interesting and challenging research direction.

## Methods
### Determining the number of repeat units deleted
The Human Genome SV Consortium (HGSVC) has created a comprehensive catalogue[11] (named HGSVC2) of the SVs in 35 samples: 34 samples from the 1000 g project (including HG00512), plus HG002 (alternatively called NA24385). Using these two catalogues, we investigate whether most CNVs in the human genome delete or duplicate whole repeat units. We determined the number of tandem repeat units deleted by the examined deletions as follows. We downloaded TRF[10]

annotations for hg38 from the UCSC Genome Browser. For each deletion, we found the tandem repeat that contains the deletion; when we found multiple such repeats, we selected the one with the shortest repeat unit. The number of deleted copies was simply obtained by dividing the length of the deletion by the length of the repeat unit.

### Building accurate alternative alleles
By building an accurate alternative allele for each CNV and aligning the reads to it, we can precisely identify reads that are supposed to support it. We downloaded PacBio HiFi reads for both HG00512 (SRR13606079 to SRR13606084) and HG002 (SRR10382244 to SRR10382249), and aligned the reads to hg38 using minimap2[29].

First, we selected a subset of CNVs so that no two CNVs were within 4000 bp of each other. Then, for a given CNV $C$, let $s$ be the start of $C$ minus 2000 bp, and $e$ be the end of $C$ plus 2000 bp. For each HiFi read overlapping $C$, we extract the portion overlapping the region from $s$ to $e$. In the case of heterozygosity, we want to identify the subset of reads supporting $C$. In order to accomplish this, for each subread, we compute two numbers, $d$ and $i$: the sum of the lengths of all deletions and insertions, respectively (≥ 10 bp in order to exclude most

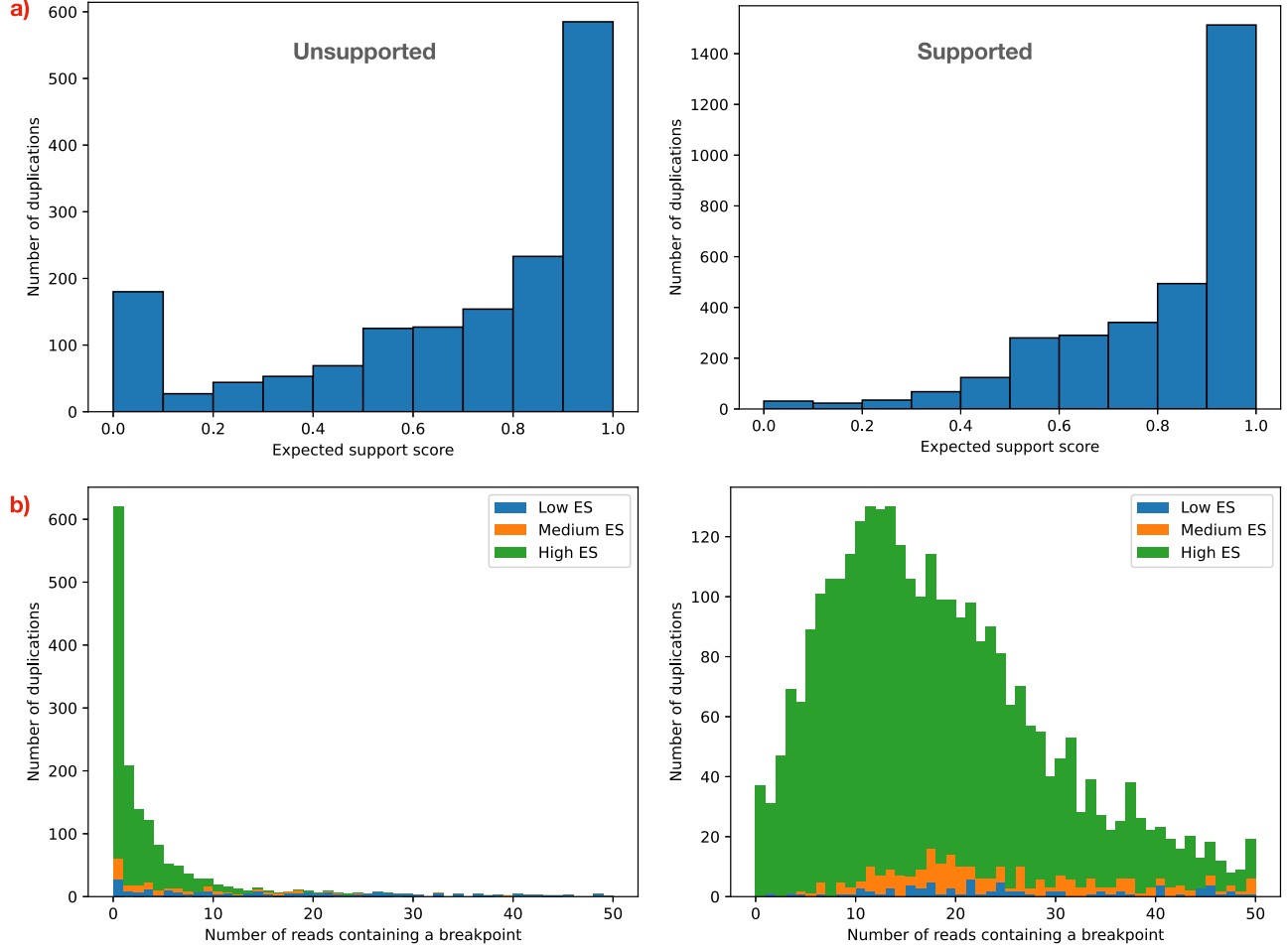

**Fig. 10 | Factors that contribute to absence of support for duplications in HG002. a** Surprisingly, only 16% of the unsupported duplications (left) have a low expected support (ES) score (< 0.3). Most unsupported duplications (77%) have high ES (≥ 0.5). Supported duplications also have high ES, as expected (right).

**b** The effect of poorly sequenced regions is even more dramatic on tandem duplications than it is on deletions. Despite the high ES, most unsupported sequences have very few or no reads containing a breakpoint.

sequencing errors). Then, we cluster the reads according to $d$ and $i$ using DBSCAN[30]. Each cluster represents an allele. In order to find the cluster that represents the allele with $C$, we find the cluster that minimises the following formula

$$\sum_{r \in S} \min_{x \in indels(r)} (size(x) - size(C))^2$$

where $S$ is a cluster of reads and $indels(r)$ is the set of indels in a read $r$. Intuitively, we want to find the cluster with the minimum sum of squared errors, where the error in a read is the difference between the length of the expected CNV and the observed one. Finally, we use the selected reads to assemble the alternative allele using SPOA[31].

Next, we aim to select a subset of alleles that are extremely likely to be correct. We apply a series of filters to our set of assembled alleles:

- **No reads**: no reads span the whole region from $s$ to $e$, therefore no alternative allele could be built;
- **No SV**: when aligning the built alternative allele to the reference, no SV ≥ 50 bp could be identified;
- **Multiple SVs**: when aligning the built alternative allele to the reference, multiple SVs ≥ 50 bp could be identified;
- **Incompatible SV**: when aligning the built alternative allele to the reference, an SV could be identified, but the length was very different from the expected CNV (size difference > 100 bp);

- **No long-read support**: when realigning the HiFi reads to the alternative allele using minimap2, we expect to find at least 3 reads that (i) cover the whole alternative allele and (ii) have no large indels (> 10bp) and a low error rate (< 0.1%). Furthermore, we use these reads to call phased variants using PEPPER-Margin-DeepVariant[9]. If the allele is correctly built, we expect one of the haplotypes to present no variant. If both haplotypes have variants, we filter the allele;
- **Other**: alternative allele was excluded for other reasons;
- **High quality**: the final set of alternative alleles retained for further analysis.

Supplementary Table 1 shows the number of assembled alleles removed in each step.

### Detecting split reads and hidden split reads supporting CNVs and expected support score
The Genome in a Bottle (GIAB) Consortium released a short-read sequencing dataset for HG002 using the Illumina HiSeq 2500 platform, and the New York Genome Centre (NYGC) released a dataset for HG00512 sequenced with the Illumina NovaSeq 6000 platform. Using these datasets, we show that many CNVs in repetitive regions are not supported by split reads, but they are supported by hidden split reads.

For each CNV for which we obtained a high quality assembly, we identify the split reads and hidden split reads supporting it as follows. We align the alternative allele to the reference genome to calculate the precise coordinates of the breakpoints of the CNV, both on the alternative allele and on the reference genome. Then, we map the short reads on the alternative allele using BWA MEM[32], and we extract all reads overlapping a breakpoint. These reads should, in theory, support the existence of the CNV when mapped to the reference genome. Let us call $S$ the set of such reads.

We extract the reference sequence $R_{ref}$ between the start and the end of the CNV, plus 2000 bp of flanking sequences in each direction. Then, we create a second sequence $R_{cnv}$ by introducing the CNV in $R_{ref}$ (i.e., deleting the deleted part or inserting the inserted sequence). Finally, we map the reads in $S$ to both $R_{ref}$ and $R_{cnv}$: The reads that are clipped when aligned to $R_{ref}$ but not when aligned to $R_{cnv}$ are split reads supporting the CNV, and reads that are not clipped and have a better alignment score when aligned to $R_{cnv}$ than when aligned to $R_{ref}$ are hidden split reads supporting the CNV.

In order to calculate the expected score (ES) for a CNV, we fix a read length $l$, and sample from the alternative allele all of the possible reads of length $l$ containing the breakpoint(s). In other words, let $b_1$ and $b_2$ be the breakpoints of the CNV on the alternative allele $A$ and let $R = \{A[i.. i + l] | i < b_1 < i + l \text{ or } i < b_2 < i + l\}$ be the set of all possible reads sequenced from the alternative allele that contains a breakpoint. Let $R'$ be the subset of reads in $R$ that aligns better to $R_{cnv}$ than to $R_{ref}$: then, $ES = \frac{|R'|}{|R|}$.

### Realigning individual hidden split reads to detect potential CNVs

In order to show that hidden split reads must be used carefully, we randomly select 10,000 repetitive regions in hg38. For each tandem repeat region, we select all reads that do not align perfectly with the reference. For each read, we split it into two at every possible location, and locally align the two half reads independently to the repetitive region. We select the split that has the highest score (calculated as the alignment score of the left half of the read plus the alignment score of the right half of the read). Let $s_l$, $e_l$ be the start and end coordinates of the optimal alignment of the left half of the read, and $s_r$, $e_r$ be the start and end coordinates of the optimal alignment of the right half of the read. If $s_r > e_l$, we detect a deletion from $e_l$ to $s_r$; vice versa, if $s_r < e_l$, we detect a duplication from $s_r$ to $e_l$. Finally, we retain all CNVs ≥ 50bp.

### Selected callers

We selected three well-performing methods from a recent benchmarking effort[7]. One requirement we enforced was that the callers would explicitly report deletions and duplications: for this reason, GRIDSS[33], another well-performing method in[7], was excluded. The callers we selected were Delly (version 1.1.3), Manta (version 1.6.0) and Smoove (version 0.2.8). We also included the original SurVIndel, because it uses a unique statistical approach that is adapted in SurVIndel2.

We ran all callers with default parameters. For Delly and Manta, we discarded calls with FILTER != PASS. Smoove and SurVIndel do not populate the FILTER field, so we retained all the calls. We also discarded all events smaller than 50 bp.

### Comparing deletions and tandem duplications

When comparing two indels, we use a set of three parameters: the *max distance*, the *min overlap* and the *maximum length difference*. Two deletions or two tandem duplications are considered to match if:

- The distance in bp between the two start coordinates and between the two end coordinates are less than or equal to the *max distance*;
- The fraction of the shortest indel that overlaps with the largest indel is at least *min overlap*;

- The difference between the lengths of the two events is less than or equal to the *maximum length difference*.

We use two sets of parameters, a *precise* (stricter) set and an *imprecise* (more permissive) set. When one or both the indels are marked as imprecise by the callers, we use the imprecise set, otherwise we use the precise set. For the precise set, we used (in order, max distance, min overlap, maximum length difference) 100 bp, 0.8, 100 bp. For the imprecise set, we used 500 bp, 0.5, 500 bp.

Compared to Truvari[34], a popular method to compare SV catalogues, we extend the comparison algorithm in two ways.

First, the same deletion or a tandem duplication in a tandem repetitive region may be represented using different coordinates. As an example, consider three genomic sequences, A, B and C, that appear as ABBBC in the reference and as ABBC in the sample (B was deleted). Any of the copies of B in the reference may be marked as deleted and it would represent the same deletion. For example, the benchmark may report the deletion of the first copy of B, while the caller reports the deletion of the last B: despite being completely disjoint, the two deletions are actually equivalent. For this reason, we employ a repeat-aware comparison method. When two deletions or two tandem duplications fall within the same tandem repeat, we only check whether the lengths of the two events are less than or equal to the maximum length difference.

Second, we extend the way insertions and tandem duplications are compared to each other. Tandem duplications in the benchmark catalogues we used are represented as insertions, i.e., as an insertion site and an inserted sequence. However, one important limitation of tandem duplications is that while insertions provide the full inserted sequence, duplications provide the duplicated sequence, and determining the accurate copy number is often very difficult with short reads. For example, consider an event where the sample sequence is ABBBC while the reference sequence is ABC. If represented as an insertion, the inserted sequence will be BB. However, if represented as a duplication, the duplicated region will be just B. The comparison method should recognise that the two represent the same event.

Our method to compare an insertion with a tandem duplication is the following. The insertion site must be within *max distance* bp of either the start or the end of the duplication. Furthermore, we check whether the inserted sequence of the insertion and the duplicated sequence of the duplication match. Let $I[1..i]$ be the inserted sequence and $D[1..d]$ be the duplicated sequence in the reference. First, we find the number of times $D$ was duplicated as $n = \lceil \frac{i}{d} \rceil$, and create $D'$ by concatenating $D$ $n$ times. Then, we compute the local alignment between $I$ and $D'$ using a scoring scheme of $+1$, $-4$, $-6$, $-1$ (match, mismatch, gap opening, gap extension). If we alignment covers at least 80% of $I$, we accept that the insertion and the duplication match.

For deletions, the sensitivity is the number of deletions in the benchmark that match any called deletion, divided by the total number of deletions in the benchmark. The precision is the number of called deletions that match any benchmark deletion, divided by the number of called deletions.

As mentioned before, tandem duplications are represented in the benchmark catalogues as insertions. Since we are interested in calculating the sensitivity only on tandem duplications, we first need to identify them from other insertions. We use the method introduced in ref. 15 to do so. Then, for duplications, the sensitivity is the number of duplications in the benchmark that match any called duplication or insertion, divided by the total number of duplications in the benchmark. The precision is the number of called duplications that match any benchmark insertion, divided by the number of called duplications.

In Supplementary Fig. 8, we show the results using the HGSVC2 benchmark for Manta, DRAGEN and SurVIndel2 (since they are the best-performing methods), using both our in-house comparison tool

and Truvari. The results largely agree, although our repeat-aware method reports slightly higher sensitivity and precision for SurVIndel2, likely due to it calling more CNVs in repetitive regions.

The comparisons performed for Fig. 7 were all performed using the imprecise parameters, to account for the possible shifting of the breakpoints due to clustering.

The code of the comparison algorithm is available at https://github.com/Mesh89/SurVClusterer.

### Clustering deletions and tandem duplications

In order to cluster a set of indels, we use the algorithm described in ref. 35. First, the set of indels is represented as a compatibility graph, where every indel is represented as a vertex, and the two indels are *compatible*, i.e., they potentially represent the same variant. We say that two indels are compatible if they match according to the comparison algorithm described in the Methods section above (we do not perform the repeat-aware comparison for efficiency reasons). After that, we heuristically compute the minimum clique cover (computing the optimal solution is not computationally feasible), and every clique is a cluster of indels, using the heuristic proposed in ref. 35. The code of the clustering algorithm was published at https://github.com/Mesh89/SurVClusterer.

### Other organisms genomic data

We downloaded Illumina paired-end and PacBio HiFi reads for seven *Arabidopsis thaliana*: Alo-0 (ERR10084935 and SRR1946106), Cas-0 (ERR10084942 and SRR1946392), Cat-0 (ERR10084948 and SRR1946393), Cvi-0 (ERR10084078 and SRR1945758), Evs-0 (ERR10084952 and SRR1946405), Hom-4 (ERR10084958 and SRR1946144) and Hum-2 (ERR10084063 and SRR1946147). All of the samples were aligned to TAIR10, using BWA MEM for short reads and minimap2 for long reads.

For the *Bos taurus* samples, we downloaded PacBio HiFi reads (ERR10378054 to ERR10378058) and two short read libraries, which we treated as independent (ERR10310239 and ERR10310240). We aligned the reads to the ARS-UCD1.3. For the *Mus musculus* sample, we downloaded PacBio HiFi reads (SRR23686163) and short reads (SRR23690179), and aligned them to the GRCm39 reference genome. For *Oryza sativa*, we downloaded PacBio HiFi reads (SRR10238608 for MH63, SRR13280199 for ZS97) and short reads (SRR13124689 for MH63, SRR13124696 for ZS97), and aligned them to the IRGSP-1.0 reference genome.

Benchmark datasets for each sample were obtained by running Sniffles2 on the long-read datasets.

### Reporting summary

Further information on research design is available in the Nature Portfolio Reporting Summary linked to this article.

## Data availability

The CNVs catalogue produced by SurVIndel2 for the 1000 g project has been deposited at EMBL-EBI European Variation Archive, project ID PRJEB71638. The IGV screenshot for manual validation of 60 reported false positive small CNVs (30 in HG002 and 30 in HG00512) have been deposited at Zenodo at: https://zenodo.org/records/10811268Data, scripts and instructions to reproduce the main figures in the paper have been deposited to GitHub in https://github.com/kensung-lab/survindel2_paper_experiments. Sequencing data for HG002 was downloaded from NCBI (accessions SRR1766442 to SRR1766486). PacBio HiFi data for HG002 was downloaded from NCBI under accession code PRJNA586863. The HG002 benchmark catalogue and the list of tier 1 regions were downloaded from https://ftp-trace.ncbi.nlm.nih.gov/ReferenceSamples/giab/data/AshkenazimTrio/analysis/NIST_SVs_Integration_v0.6/. Information on accessing the 3202 CRAM

files for the 1KGP project produced by NYGC can be found at https://www.internationalgenome.org/data-portal/data-collection/30x-grch38. The Phase 2 benchmark calls produced by HGSVC are available at https://www.internationalgenome. org/data-portal/data-collection/hgsvc2. The SV catalogue produced by NYGC was downloaded from http://ftp.1000genomes.ebi.ac.uk/vol1/ftp/data_collections/1000G_2504_high_coverage/working/20210124.SV_Illumina_Integration/1KGP_3202.gatksv_svtools_novelins.freeze_V3.wAF.vcf.gz. Data from the 1001 Genomes Project was downloaded from NCBI under accession code PRJNA273563. Accession codes for individual samples are: Alo-0 (ERR10084935 and SRR1946106), Cas-0 (ERR10084942 and SRR1946392), Cat-0 (ERR10084948 and SRR1946393), Cvi-0 (ERR10084078 and SRR1945758), Evs-0 (ERR10084952 and SRR1946405), Hom-4 (ERR10084958 and SRR1946144) and Hum-2 (ERR10084063 and SRR1946147). All the data for the other organisms were downloaded from NCBI, and accessions will be provided in the following paragraphs. PacBio HiFi reads for the *Bos taurus* samples were downloaded from accessions ERR10378054 to ERR10378058, while two short read libraries, which we treated as independent samples, were also downloaded from accessions ERR10310239 and ERR10310240. PacBio HiFi reads for the *Mus musculus* sample was downloaded from accession SRR23686163, while short reads were from accession SRR23690179. PacBio HiFi reads for the *Oryza sativa* samples were downloaded from accession SRR10238608 (for MH63) and SRR13280199 (for ZS97), while short reads from accession SRR13124689 (for MH63) and SRR13124696 (for ZS97).

## Code availability

The code for SurVIndel2 is available at https://github.com/kensung-lab/SurVIndel2/. The code and instructions to reproduce the main figures in the manuscript can be found at https://github.com/kensung-lab/survindel2_paper_experiments/.

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

## Acknowledgements

This work was supported by JSPS KAKENHI Grant Number JP23KF0205, JSPS Postdoctoral Fellowships for Research in Japan and the Hong Kong Jockey Club Charities Trust to the JC STEM Lab of Computational Genomics.

## Author contributions

R.R. developed and implemented the method, performed the benchmarking and conducted the analyses. W.K. helped develop the method and provided guidance. R.R. and W.K. wrote the manuscript.

## Competing interests

The authors declare no competing interests.
