## [Peer Review File · Nature Communications]

SurVIndel2: improving copy number variant calling from next-generation sequencing using hidden split readsREVIEWER COMMENTS

Reviewer #1 (Remarks to the Author):

The authors describe surVIndel2, a tool for CNV detection, leveraging a previously underappreciated signal - the so-called hidden split reads. This innovative approach appears successful in identifying CNVs, especially in a tandem repeat context, where typical split reads fail to identify the presence of a CNV. The observation that many CNVs are missed due to poor coverage is particularly intriguing, and I wonder if sequence-specific features beyond repetitive context contribute to this. The performance of this tool, compared to well-established tools, is impressive, and I wish to congratulate the authors on their work.

Sincerely,
Wouter De Coster

Major:

I see the authors use SurVClusterer to compare SVs from various callsets, but the manuscript could be improved by more information on how exactly that is done, and how the precision and sensitivity are calculated based on that. Note that a commonly accepted method for comparing structural variant call sets is Truvari (<https://github.com/ACEnglish/truvari>). Can the authors confirm that they obtained similar results with their tool?

Minor:

It appears the authors use the 'HSR' abbreviation and only describe what it means later in the text.

The y-axis in Figure 1c and 1d is unlabeled.

There is no legend for the bar colors in Figure 5b, but presumably, the colors are the same as for 5a. However, this is not explicitly stated and could lead to confusion.

Mostly out of curiosity and not crucial for this paper, I wondered if it is possible to train the filter module on a combination of different species and what the performance of such a mixed model would be compared to the species-specific filter.

Reviewer #2 (Remarks to the Author):

This work presents a novel approach, SurVIndel2, to detect copy number variations (CNVs). It looks for sequences that can be aligned to alternative positions to the reference genome as split reads (referred to as hidden split read) and demonstrates enhanced sensitivity and precision from short-read sequencing data, particularly improving detection of duplication in repetitive regions of the genome.

Although long-read sequencing can significantly improve CNV calling in repetitive regions, this work acknowledges that short-read sequencing will remain the primary research data in the near future and therefore addressing the limitation of detecting CNVs in repetitive regions using short-read sequencing is of great relevance.

The manuscript provides an evaluation of SurVIndel2 with public datasets of multiple organisms and benchmarks its performance with existing CNV and SV callers. However, further analyses and validation on existing tools could strengthen the claims. Please see the list of revisions below for details.

The analysis appears methodologically sound overall, but a more detailed discussion on limitations and

potential biases would be beneficial. Any flaws do not seem to prohibit publication but may require revisions (see the list of revisions below) for clarity and completeness.

The manuscript clearly described the methodology and provides multiple benchmarking analyses of the method on cell line samples. I believe the work meets current standards in bioinformatics for CNV detection, given the list of revisions are addressed.

While SurVIndel2 is open-source software, for full reproducibility, I suggest also including scripts for running the analyses.

List of Revisions:

1. Figure 2 shows that over 93% of the tandem duplications of HG002 are in repeat regions. It would be good to see a breakdown of the types of repeat regions in the bar chart.
2. In section 2.3, the manuscript stated that filtering was implemented by leave-1-out training for SurVIndel2, but methods of the 'pre-trained random forest' filtering are not provided. Please include further description.
3. In addition to revision 2, section 2.3 didn't include descriptions of how filtering was performed on CNV calls of benchmarking tools. For example, Manta ranks the quality of SV calls using QUAL score and FILTER tags, and the choice of filtering strategy can impact the benchmarking results, particularly sensitivity. Figure 4 was generated with one set of calls per sample per caller. Possibilities of alternative filtering strategies were not explored. I recommend including multiple filtering options in the benchmarking, such as different QUAL cutoffs per sample per caller, and plot curves of precision-recall in figure 4(a), 4(b), and 5(a).
4. HG002 is a great independent sample, which was not specifically trained on by SurVIndel2, for benchmarking purposes. However, the discussion only includes sensitivity, not precision. I strongly recommend including HG002 in Figure 4(a) and 4(b) as well.
5. Very importantly, in section 2.3, Delly, Lumpy, and Manta were selected as benchmarking tools according to reference [4]. However, in this benchmarking study, GRIDSS is another top-performing SV caller. I strongly recommend including benchmarking with GRIDSS2 + Purple (Cameron, D.L., Baber, J., Shale, C. et al. GRIDSS2: comprehensive characterization of somatic structural variation using single breakend variants and structural variant phasing. *Genome Biol* 22, 202 (2021). <https://doi.org/10.1186/s13059-021-02423-x>).
6. It's interesting to see that SurVIndel 1&2 are more sensitive than other callers with no HSR support. Section 2.7 discusses the reasons why these CNVs have a low level of evidence. It would be good to include more discussion on why SurVIndel2 can capture more such CNVs than other callers, as well as the true positive rate of unsupported CNVs called by SurVIndel2.
7. I recommend including a discussion on the limitations of the tool regarding the detection of somatic CNVs in cancer genomes. This discussion could acknowledge that, unlike the other tools benchmarked in the study, SurVIndel2 does not currently support the identification of somatic CNVs, which are critical for understanding cancer genomics. Highlight this as a significant caveat, and suggest future research directions or potential updates to the tool that could address this gap. Emphasize the importance of this functionality for comprehensive cancer genomics studies and how its inclusion could further elevate the tool's applicability and utility in the field. This addition will not only provide a more balanced view of SurVIndel2's capabilities but also align expectations for researchers interested in cancer genomics.

Reviewer #3 (Remarks to the Author):

This manuscript introduces a new tool SurVIndel2 for detecting the copy number variations (CNV) using short illumina sequencing reads with a novel technique that uses hidden split reads and machine learning techniques. The benchmarking results showed that SurVIndel2 performed better than other

short-read-based structural variant (SV) callers. However, concerns arise regarding the benchmarking evaluation methods and the availability of results produced in this study.

Major issues:

1. Throughout the manuscript, it seems the authors have considered all deletion (≥ 50 bp) and tandem duplication type SVs as CNVs. There is confusion as all deletions (≥ 50 bp) can not be called CNV. Is the tool designed to detect all DEL and DUP-type SVs? If that is the case, then it is not suggested to call it a CNV caller tool.

2. The authors have claimed that it is a new short-read-based CNV caller, however, in the last few steps it uses the machine learning technique that takes advantage of long-read-based CNV callers (steps g and h) for training purposes. The machine learning model of SurVIndel2 is used in these steps, right? So would it be fair to call it a short-read-based caller? It is expected to work a little better when long-reads or long-read-based callsets are used at the filtering stage, so it won't be a fair comparison with other "pure" short-read-based callers such as Manta, Lumpy, and Delly. Also, Lumpy only calls the DEL-type SVs, so it may not detect any INS (or DUP) type SVs.

3. The "Method" section is included inside the "Result" section. Also, that section only talks about the algorithms and techniques used in this tool. It would be better to include the benchmarking strategies such as what comparison tools and the truth set are used for evaluation. The results can be reproduced for verification if the VCF file for the truth sets, callsets, commands, or scripts are available in the "Methods" section. The authors could provide a link (e.g. GitHub repo) with all the result files.

4. The authors have used the HGSVC2 callsets as truth sets, so it is not clear if only DEL and DUP type SVs or the whole SV set is selected for the evaluations. Are these high-confidence call sets? Are they validated against GIAB truth sets? For each sample, the evaluation was performed using a training module of SurVIndel which was trained by using the features of other 33 long-read-based CNV callsets. So would it be also possible to compare the results of SurVIndel2 without using the last step to make a fair comparison with "pure" short-read-based callers? The evidence and claims can be better examined with the availability of callsets. Another suggestion is to use the GIAB SV truth set (v0.6) can also be used for the evaluation. The DEL-type SVs with length ≥ 1 kbp can be used as a benchmark set while evaluating all the callers. Also, why the precision is not computed for DEL types (last line of 1st paragraph of Section 2.3)?

5. All the other tools that are used in this comparison are not designed to call CNVs. Another most commonly used CNV caller tool is CNVnator. So it is suggested to use that tool to compare the performance of SurVIndel2. Also, the 2nd best-performing tool, Manta, has been improved in the new DRAGEN pipeline released by Illumina. The authors are suggested to take a look at the pre-print (<https://doi.org/10.1101/2024.01.02.573821>) and the available callsets as some of the evaluation numbers presented in the pre-print for HG002 do not match the analysis done in this manuscript.

6. The authors used the terms like "tandem repetitive regions", and "repetitive regions" throughout the manuscript. It would be better if they are mentioned or a BED file is provided to identify those regions in reference. Are they telomere or centromere regions? It is difficult to understand the CNVs outside repetitive regions if they are not clearly defined or explained. Also, it is better to use the length information while using "shorter" terms for CNVs.

We thank the reviewers for the useful feedback. As requested, we have uploaded instructions and code to reproduce the figures of the paper in https://github.com/kensung-lab/survindel2_paper_experiments/
We have also updated the data availability, since the data is now available in EBI.
We will reply to their reviews point by point:

Reviewer 1

The authors describe `surVIndel2`, a tool for CNV detection, leveraging a previously underappreciated signal - the so-called hidden split reads. This innovative approach appears successful in identifying CNVs, especially in a tandem repeat context, where typical split reads fail to identify the presence of a CNV. The observation that many CNVs are missed due to poor coverage is particularly intriguing, and I wonder if sequence-specific features beyond repetitive context contribute to this. The performance of this tool, compared to well-established tools, is impressive, and I wish to congratulate the authors on their work.

Sincerely,
Wouter De Coster

We appreciate the kind words.

Regarding whether there are sequence-specific features that can explain low coverage, it would be a very interesting research direction. We have observed that many of these regions tend to be very repetitive, e.g., low complexity sequences or having long homopolymers. However, we have also observed pairs of very similar low complexity regions in totally different locations where one region was well sequenced while the other was very poorly sequenced. Therefore, there must be factors other than the sequence itself.

We would like to emphasize that we have not conducted any rigorous study on this at this point, and these are just simple observations made while developing the tool.

Major:

I see the authors use `SurVClusterer` to compare SVs from various callsets, but the manuscript could be improved by more information on how exactly that is done, and how the precision and sensitivity are calculated based on that. Note that a commonly accepted method for comparing structural variant call sets is `Truvari` (<https://github.com/ACEnglish/truvari>). Can the authors confirm that they obtained similar results with their tool?

The main reason why we use `SurVClusterer` for comparing SVs is that it employs a more sophisticated algorithm for comparing insertions and duplications. In long reads-based benchmarks such as `HGSVC2`, duplications are reported as point insertions. Short read based tools often report them as tandem duplications, denoted with the start and the end of the duplicated region.

One issue of such representation is that we do not know how many times the region was duplicated (the caller may report an estimate, but it is generally difficult to estimate accurately for duplications that are not extremely long, i.e., 1000s of bps).

Therefore, if a tool reports that a region B was duplicated, and the benchmark reports an insertion of a sequence BBBB in the same location, the comparison algorithm in `SurVClusterer` identifies that the two represent the same event. On the other hand, it appears to us (by reading the algorithm explained in the documentation) that `truvari` assumes every tandem duplication is duplicated exactly once.

One clear example of this is the benchmark SV `chr6-168229517-INS-512` in `NA24385`. `surVIndel2` predicts a duplication of 128 bp, and the inserted sequence reported by the benchmark is identical to the duplicated region predicted by `surVIndel2`, duplicated 4 times. `SurVClusterer` reports the two events as matching, while `truvari` does not.

In practice, many cases are less clear-cut; for example, many tandem duplications are caused by expansions of short motifs. When such expansions are not much smaller than the read length, it is nearly impossible to identify the exact length of the expansion using short reads (especially given that many of them are in the very noisy regions that we describe in Section 2.7), and it is often underestimated when compared to HiFi reads.

Secondly, SurVClusterer uses the TRF annotations to determine whether two events within the same repetitive region represent the same event but shifted.

In any case, we did try using truvari to compare Manta and SurVIndel2 (since they perform far better than the rest). Instructions and data are reported in https://github.com/kensung-lab/survindel2_paper_experiments/, under truvari.

For deletions, we obtained the following results:

HG00512

Manta: 0.36 recall, 0.72 precision
SurVIndel2: 0.56 recall, 0.80 precision

HG002

Manta: 0.36 recall, 0.71 precision
SurVIndel2: 0.56 recall, 0.75 precision

Since the matching criteria of the two comparison algorithms are different, the numbers are a bit different; however, the relative performance are the same.

For tandem duplications, because of the reasons explained above, we have disabled the comparison of the inserted sequences. If activated, the precision of the algorithms drop to around 0.6.

HG00512

Manta: 0.13 recall, 0.78 precision
SurVIndel2: 0.37 recall, 0.83 precision
HG002

Manta: 0.14 recall, 0.77 precision
SurVIndel2: 0.41 recall, 0.82 precision

However, this comparison is perhaps too relaxed, and we believe the one operated by SurVClusterer is closer to reality.

It must be noted that comparing SVs is very challenging, due to repetitive regions, imprecise breakpoints, imprecise inserted sequences, multiple valid representations for the same event, etc..

We are always looking for improvements and cases where our algorithm fails. Aside from benchmarking, being able to correctly identify matching SVs is especially important when clustering SVs in a population.

We also added a paragraph to the supplementary section "Comparing deletions and tandem duplications" stating how recall and precision are calculated.

Minor:

It appears the authors use the 'HSR' abbreviation and only describe what it means later in the text.

Indeed, we never introduced the HSR abbreviation formally. We added it to Section 2.1.

The y-axis in Figure 1c and 1d is unlabeled.

Since Figure 1 has no plot with x and y axes in it, we wonder if the reviewer is referring to a different figure. We have indeed found some panels without y-axis label:

- Fig. 2f and S6f
- Fig. 4c and d

- Fig. S13c (now S15c) and S14c (now S16c)
We added a label to all of them.

There is no legend for the bar colors in Figure 5b, but presumably, the colors are the same as for 5a. However, this is not explicitly stated and could lead to confusion.

We now explicitly state that the legend of panel (a) also applies to (b).

Mostly out of curiosity and not crucial for this paper, I wondered if it is possible to train the filter module on a combination of different species and what the performance of such a mixed model would be compared to the species-specific filter.

It is not easy to answer this question, as there are currently few (that we could find) datasets having both HiFi and Illumina reads. The reason we need HiFi is that we need to divide our training data into false and true positives. To do so, we require a benchmark that reports accurate inserted sequences. At the time of our experiments, we found that long read SV callers only produced accurate inserted sequences with HiFi reads.

Out of curiosity, we tried training a model on both humans and Arabidopsis data, and tried to predict an Arabidopsis sample (which was left out of training). The results are similar to using Arabidopsis data only, perhaps slightly worse: compared to using Arabidopsis only, for deletions, the mixed model has 0.02 lower sensitivity and 0.01 higher precision. For duplications, the mixed model has 0.01 higher sensitivity but 0.05 lower precision. We also tried to classify a different organism (cattle), and there is no appreciable difference between the human-only model and the mixed model (very tiny increase in predicted TPs, but not enough to generate a change in sensitivity). It is possible that a model that is larger or more sophisticated could utilize the mixed training model better.

Reviewer 2

This work presents a novel approach, SurVIndel2, to detect copy number variations (CNVs). It looks for sequences that can be aligned to alternative positions to the reference genome as split reads (referred to as hidden split read) and demonstrates enhanced sensitivity and precision from short-read sequencing data, particularly improving detection of duplication in repetitive regions of the genome.

Although long-read sequencing can significantly improve CNV calling in repetitive regions, this work acknowledges that short-read sequencing will remain the primary research data in the near future and therefore addressing the limitation of detecting CNVs in repetitive regions using short-read sequencing is of great relevance.

The manuscript provides an evaluation of SurVIndel2 with public datasets of multiple organisms and benchmarks its performance with existing CNV and SV callers. However, further analyses and validation on existing tools could strengthen the claims. Please see the list of revisions below for details.

The analysis appears methodologically sound overall, but a more detailed discussion on limitations and potential biases would be beneficial. Any flaws do not seem to prohibit publication but may require revisions (see the list of revisions below) for clarity and completeness.

The manuscript clearly described the methodology and provides multiple benchmarking analyses of the method on cell line samples. I believe the work meets current standards in bioinformatics for CNV detection, given the list of revisions are addressed.

While SurVIndel2 is open-source software, for full reproducibility, I suggest also including scripts for running the analyses.

We would like to thank the reviewer. As suggested, we have published the scripts for generating the main figures in https://github.com/kensung-lab/survindel2_paper_experiments/.

Below, we will reply to the individual revisions.

List of Revisions:

1. Figure 2 shows that over 93% of the tandem duplications of HG002 are in repeat regions. It would be good to see a breakdown of the types of repeat regions in the bar chart.

We use TRF to identify tandem repetitive regions (Supplementary). We have added a sentence in Section 2.1 that explicitly states this. As far as we know, TRF does not classify repetitive regions into subtypes, and I am currently not aware of such a classification (as opposed as general repeats, which can be classified into mobile elements, tandem repetitive regions, etc.).

2. In section 2.3, the manuscript stated that filtering was implemented by leave-1-out training for SurVIndel2, but methods of the 'pre-trained random forest' filtering are not provided. Please include further description.

We added a new Supplementary section called "Filtering module training".

3. In addition to revision 2, section 2.3 didn't include descriptions of how filtering was performed on CNV calls of benchmarking tools. For example, Manta ranks the quality of SV calls using QUAL score and FILTER tags, and the choice of filtering strategy can impact the benchmarking results, particularly sensitivity. Figure 4 was generated with one set of calls per sample per caller. Possibilities of alternative filtering strategies were not explored. I recommend including multiple filtering options in the benchmarking, such as different QUAL cutoffs per sample per caller, and plot curves of precision-recall in figure 4(a), 4(b), and 5(a).

We retained calls with FILTER == PASS, since this is in our opinion the standard use case. We added a Supplementary section called "Selected callers" where we specify this.

We have also generated a precision/recall curve for deletions in HG00512 and HG002 (Smoove, SurVIndel and SurVIndel2 do not report a QUAL value and therefore they are just single points).

We could not generate a curve for duplications because for Delly and Manta we use two different call sets to calculate recall and precision: precision is calculated on SVTYPE=DUP, while for recall we also include SVTYPE=INS (since Delly and Manta may decide to report events as either DUP or INS).

The code and data for generating the curves was uploaded to the aforementioned GitHub repository.

4. HG002 is a great independent sample, which was not specifically trained on by SurVIndel2, for benchmarking purposes. However, the discussion only includes sensitivity, not precision. I strongly recommend including HG002 in Figure 4(a) and 4(b) as well.

HG002 is part of the HGSVC2, as mentioned in Section 2.3: *a catalogue (called HGSVC2) of insertions and deletions in 35 human genomes (34 from the 1000 genomes project, plus HG002)*. In HGSVC2, it is called NA24385. Therefore, it is in Figure 4a and 4b. The purpose of figures 4c and 4d is simply to show that existing callers struggle with HSR-supported calls.

We have added two more benchmarks for HG002 (Fig. S7 and an additional paragraph in Section 2.3). One was the GIAB v0.6 benchmark, while the other was obtained by running Sniffles2 on HiFi reads. The relative performance of the tools are in line with what was observed on the HGSVC2 benchmark.

Instructions and files to reproduce the benchmark are in the GitHub repository.

5. Very importantly, in section 2.3, Delly, Lumpy, and Manta were selected as benchmarking tools according to reference [4]. However, in this benchmarking study, GRIDSS is another top-performing SV caller. I strongly recommend including benchmarking with GRIDSS2 + Purple (Cameron, D.L., Baber, J., Shale, C. et al. GRIDSS2: comprehensive characterization of somatic

structural variation using single breakend variants and structural variant phasing. *Genome Biol* 22, 202 (2021). <https://doi.org/10.1186/s13059-021-02423-x>.

We have tried using GRIDSS, but it does not explicitly report deletions and duplications, only breakends. One criteria we used is that the callers should explicitly predict deletions and tandem duplications.

In the GitHub README, the authors mention that `example/simple-event-annotation.R` will annotate the calls with `SIMPLE_TYPE` and `SVLEN`, but the script is not made for general users as it has hardcoded paths inside. We modified the hardcoded path to point to our own GRIDSS results, but the script printed this:

```
Error in .subassign_columns(x, nsbs, value) :  
  provided 68 variables to replace 67 variables  
Calls: [  
In addition: Warning message:  
info fields with no header: SVLEN  
Execution halted
```

In the benchmarking paper “Comprehensive evaluation and characterisation of short read general-purpose structural variant calling software” by Cameron DL et al. (the authors of GRIDSS), it is shown that GRIDSS performs similarly to Manta on HG002 (Fig.1, slightly more precise but less sensitive).

6. It's interesting to see that SurVIndel 1&2 are more sensitive than other callers with no HSR support. Section 2.7 discusses the reasons why these CNVs have a low level of evidence. It would be good to include more discussion on why SurVIndel2 can capture more such CNVs than other callers, as well as the true positive rate of unsupported CNVs called by SurVIndel2.

We investigated why SurVIndel2 can detect a portion of deletions marked as unsupported. There seem to be many factors contributing.

We found that 38% of these deletions are predicted using HSRs. As for why such deletions are classified as unsupported despite having HSRs supporting them, we identified three possible reasons:

- (1) `spoa` built an alternative allele that is not 100% correct, and it escaped our filters;
- (2) the HSRs are less than 5 (minimum requirement to be classified as HSR-supported), while SurVIndel2 is able to identify CNVs from as few as 3 reads;
- (3) when mapping the short reads to the set of alternative alleles, BWA might have placed the relevant reads somewhere else (since we mapped the reads to all of the alternative alleles together, as running BWA for thousands of alternative allele separately would have been unfeasible with our computational resources).

Another 38% of them were predicted using the discordant pairs module, so no SR or HSR support was required. Finally, out of the remaining 24% that were predicted using split reads, nearly half of them had less than 5 split reads supporting them (minimum support required to be classified as SR-supported was again 5).

We have added a small paragraph explaining this to Section 2.3.

7. I recommend including a discussion on the limitations of the tool regarding the detection of somatic CNVs in cancer genomes. This discussion could acknowledge that, unlike the other tools benchmarked in the study, SurVIndel2 does not currently support the identification of somatic CNVs, which are critical for understanding cancer genomics. Highlight this as a significant caveat, and suggest future research directions or potential updates to the tool that could address this gap. Emphasize the importance of this functionality for comprehensive cancer genomics studies and how its inclusion could further elevate the tool's applicability and utility in the field. This addition will not only provide a more balanced view of SurVIndel2's capabilities but also align expectations for researchers interested in cancer genomics.

Indeed, SurVIndel2 does not explicitly support detecting somatic variants (excluding the naive approach of processing tumor and normal independently and operating a subtraction). I have added a paragraph to the Discussion explaining this.

Reviewer 3

This manuscript introduces a new tool SurVIndel2 for detecting the copy number variations (CNV) using short illumina sequencing reads with a novel technique that uses hidden split reads and machine learning techniques. The benchmarking results showed that SurVIndel2 performed better than other short-read-based structural variant (SV) callers. However, concerns arise regarding the benchmarking evaluation methods and the availability of results produced in this study.

We would like to thank the reviewer. Below, we reply to their comments individually:

Major issues:

1. Throughout the manuscript, it seems the authors have considered all deletion (≥ 50 bp) and tandem duplication type SVs as CNVs. There is confusion as all deletions (≥ 50 bp) can not be called CNV. Is the tool designed to detect all DEL and DUP-type SVs? If that is the case, then it is not suggested to call it a CNV caller tool.

By reading the literature, it appears to us that there is no universally accepted definition of CNV.

Some definitions require a length of at least 1000 bp, other 50 bp:

<https://www.ncbi.nlm.nih.gov/pmc/articles/PMC9407502/>

<https://genomemedicine.biomedcentral.com/articles/10.1186/s13073-021-00945-4>

<https://www.ncbi.nlm.nih.gov/pmc/articles/PMC9365719/>

However, it is true that we do not detect all CNVs, because we only detect tandem duplications (as opposed to interspersed duplications). In the original SurVIndel, we created the term *local CNV* to refer to deletions and tandem duplications only. We have clarified in the introduction that in the paper we consider local CNVs, and we will refer to them simply as CNV for convenience.

2. The authors have claimed that it is a new short-read-based CNV caller, however, in the last few steps it uses the machine learning technique that takes advantage of long-read-based CNV callers (steps g and h) for training purposes. The machine learning model of SurVIndel2 is used in these steps, right? So would it be fair to call it a short-read-based caller? It is expected to work a little better when long-reads or long-read-based callsets are used at the filtering stage, so it won't be a fair comparison with other "pure" short-read-based callers such as Manta, Lumpy, and Delly. Also, Lumpy only calls the DEL-type SVs, so it may not detect any INS (or DUP) type SVs.

We distribute the data used to train the model (as the trained model is several GBs and difficult to distribute, and training the model requires modest computational resources). It is true that the data was obtained with the help of long reads. However, as mentioned, no long reads are required to run SurVIndel2, only short reads, and for this reason we believe it to be a short reads caller. It is true that if the user wants to train its model on different samples (for example, a different organism) he will need a high quality ground truth set for those samples, which are usually obtained by long reads. However, we have demonstrated in the paper that the default model provided works well in several non-human organisms.

Lumpy calls deletions, duplications, inversions and generic "breakends".

Regarding the fairness of the comparison, we only want to demonstrate that the tool will be useful to the community. Delly, Lumpy and Manta are seminal works that have been developed nearly a decade ago. We have been able to develop our algorithm not only thanks to them, but also thanks to data that has been published since then, such as the many long reads datasets, novel catalogues such as HGVC2 that are more complete than ever, and even high quality assemblies such as HG002. We have developed our insights and ultimately our algorithm using this data, which was not available at the time the other tools were developed.

3. The "Method" section is included inside the "Result" section. Also, that section only talks about the algorithms and techniques used in this tool. It would be better to include the benchmarking strategies such as what comparison tools and the truth set are used for evaluation. The results can be reproduced for verification if the VCF file for the truth sets, callsets, commands, or scripts are available in the "Methods" section. The authors could provide a link (e.g. GitHub repo) with all the result files.

4. The authors have used the HGVC2 callsets as truth sets, so it is not clear if only DEL and DUP type SVs or the whole SV set is selected for the evaluations. Are these high-confidence call sets? Are they validated against GIAB truth sets? For each sample, the evaluation was performed using a training module of SurVIndel which was trained by using the features of other 33 long-read-based CNV callsets. So would it be also possible to compare the results of SurVIndel2 without using the last step to make a fair comparison with “pure” short-read-based callers? The evidence and claims can be better examined with the availability of callsets. Another suggestion is to use the GIAB SV truth set (v0.6) can also be used for the evaluation. The DEL-type SVs with length ≥ 1 kbp can be used as a benchmark set while evaluating all the callers. Also, why the precision is not computed for DEL types (last line of 1st paragraph of Section 2.3)?

We will respond to both points together.

We understand this is confusing, since we failed to mention it explicitly. Tandem duplications in the HGVC2 benchmark, but also in the Sniffles2 results, are reported as insertions. We have a method that we have previously published in the Supplementary of <https://www.nature.com/articles/s41467-023-38870-2> that identifies which insertions are due to tandem duplication. We tried to make it clearer by explicitly mentioning this in 2.3 and 2.4, and also clarifying how sensitivity and precision are calculated in the supplementary section “Comparing deletions and tandem duplications”. We only use tandem duplications to calculate sensitivity, while we use all benchmark insertions to calculate precision.

We have published all the instructions, data and code to replicate our figures in https://github.com/kensung-lab/survindel2_paper_experiments/.

We believe HGVC2 to be high-confidence, since it is produced by a well known consortium (the Human Genome SV Consortium) using HiFi reads. It is generally more complete than the GIAB benchmark (although, in our experience, it contains more false positives). They are hard to compare since GIAB is, as far as we know, hg19 only.

In any case, we have added two more benchmarks for HG002: the GIAB one, and one generated by Sniffles2 (Fig. S7). Rather than ≥ 1000 bp, we used ≥ 50 bp, since all the tools involved are designed to call SV-type events.

Unfortunately the unfiltered callset of SurVIndel2 is very noisy, as it contains $>100,000$ calls, and it is certainly not recommended to use it directly. For this reason, it is not meaningful to benchmark it.

Precision was computed for all samples, including HG002 (which is the sample NA24385 in HGVC2). It is the y-axis in Fig. 4a and 4b. As mentioned, we have added two more benchmarks for HG002, and we computed sensitivity and precision.

5. All the other tools that are used in this comparison are not designed to call CNVs. Another most commonly used CNV caller tool is CNVnator. So it is suggested to use that tool to compare the performance of SurVIndel2. Also, the 2nd best-performing tool, Manta, has been improved in the new DRAGEN pipeline released by Illumina. The authors are suggested to take a look at the pre-print (<https://doi.org/10.1101/2024.01.02.573821>) and the available callsets as some of the evaluation numbers presented in the pre-print for HG002 do not match the analysis done in this manuscript.

In the original SurVIndel paper, we did include CNVnator, and it performed much worse than the callers included here: <https://academic.oup.com/bioinformatics/article/37/11/1497/5466452>. Also, the scope of CNVnator is different, as it is designed to detect very large CNVs, while SurVIndel2 and the other callers tested are designed to detect ≥ 50 bp events (or even smaller).

We did not compare DRAGEN to free and open-source because it is proprietary, closed and requires a license. However, we recognize its prominence in the field and since the paper the reviewer mentions does provide calls for 1KGP, including the samples in HGVC2, we included a comparison (Section 2.3 and Fig. S8). DRAGEN is a very clear improvement over Manta, but its sensitivity and recall are still sensibly lower than SurVIndel2. As usual, the GitHub repository contains code and data for this comparison.

6. The authors used the terms like “tandem repetitive regions”, and “repetitive regions” throughout the manuscript. It would be better if they are mentioned or a BED file is provided to identify those regions in reference. Are they telomere or centromere regions? It is difficult to understand the CNVs outside repetitive regions if they are not clearly defined or explained. Also, it is better to use the length information while using “shorter” terms for CNVs.

We now specify in Section 2.1 that we consider tandem repetitive regions identified by the Tandem Repeats Finder (TRF). In the GitHub we provide a file with the list of repetitive regions.

Sincerely,

Ramesh Rajaby and Wing-Kin Sung.

REVIEWER COMMENTS

Reviewer #1 (Remarks to the Author):

Dear authors,

I am satisfied with the clarifications made regarding my previous request, and wish to congratulate you on your manuscript and tool.
I have no further comments.

Sincerely,
Wouter De Coster

Reviewer #2 (Remarks to the Author):

The authors have addressed my concerns and suggestions raised in my previous review. The revisions made to the manuscript and supplementary methods, including the clarified analyses and expanded discussions, have improved the clarity of the study. The additional instructions and code provided transparency and reproducibility of the published results. Therefore, I recommend this manuscript for publication.

Reviewer #2 (Remarks on code availability):

The repository for SurVIndel2 (<https://github.com/kensung-lab/SurVIndel2>) includes a clear README file with installation and running instructions. I successfully installed SurVIndel2 by following these instructions on Rocky Linux v8.9 (Green Obsidian).

The repository also provides clear instructions for reproducing the main figures. However, due to time constraints, I have not attempted to reproduce the results presented in the manuscript.

Reviewer #3 (Remarks to the Author):

I would like to thank the authors for putting a great effort and doing additional analysis as suggested by the reviewers. They have indeed provided more explanations for most of the comments and suggestions.

However, there are still a few additional concerns that may need to be addressed.

In the earlier version of the manuscript, it was a bit unclear about the validation tool/software that was used for all the analysis, and Reviewer#1 suggested using Truvari. I think Truvari has been considered a standard tool for SV validation. Also, some recent results have used Wittyer (<https://github.com/Illumina/witty.er>) for this purpose. The authors have clarified that they have used a tool "SurVClusterer" which was developed by them and also clarified some reasons why it may be better than Truvari. They have also provided some additional results based on truvari. I have tried to replicate some of the results by running truvari. Here are some comments based on that result.

1. The file "1_data_preparation/by-sample/HG00512.DEL.vcf.gz" was not present on their Github repo (https://github.com/kensung-lab/survindel2_paper_experiments/). So it would be great if they upload at least DEL benchmark results for few samples.

2. For HG002 with hg19 reference calls, truvari failed with error "ERROR:root:No SAMPLE columns found in vcf", though it worked for Manta and DRAGEN vcfs.

I think it would be great if authors could show that the performance of their tool is better than Manta/DRAGEN (and others) by running Truvari on the DEL variant types.

Reviewer #3 (Remarks on code availability):

The codes are reproducible, but some files are still missing and also some info inside VCFs is missing.

We thank the reviewers for the kind feedback. We will reply to their reviews point by point:

Reviewer 1

Dear authors,

I am satisfied with the clarifications made regarding my previous request, and wish to congratulate you on your manuscript and tool.
I have no further comments.

Sincerely,
Wouter De Coster

Thank you very much.

Reviewer 2

The authors have addressed my concerns and suggestions raised in my previous review. The revisions made to the manuscript and supplementary methods, including the clarified analyses and expanded discussions, have improved the clarity of the study. The additional instructions and code provided transparency and reproducibility of the published results. Therefore, I recommend this manuscript for publication.

The repository for SurVIndel2 (<https://github.com/kensung-lab/SurVIndel2>) includes a clear README file with installation and running instructions. I successfully installed SurVIndel2 by following these instructions on Rocky Linux v8.9 (Green Obsidian).

The repository also provides clear instructions for reproducing the main figures. However, due to time constraints, I have not attempted to reproduce the results presented in the manuscript.

Thank you very much.

Reviewer 3

I would like to thank the authors for putting a great effort and doing additional analysis as suggested by the reviewers. They have indeed provided more explanations for most of the comments and suggestions.

However, there are still a few additional concerns that may need to be addressed.

In the earlier version of the manuscript, it was a bit unclear about the validation tool/software that was used for all the analysis, and Reviewer#1 suggested using Truvari. I think Truvari has been considered a standard tool for SV validation. Also, some recent results have used Wittyer (<https://github.com/Illumina/witty.er>) for this purpose. The authors have clarified that they have used a tool "SurVClusterer" which was developed by them and also clarified some reasons why it may be better than Truvari. They have also provided some additional results based on truvari. I have tried to replicate some of the results by running truvari. Here are some comments based on that result.

1. The file "1_data_preparation/by-sample/HG00512.DEL.vcf.gz" was not present on their Github repo (https://github.com/kensung-lab/survindel2_paper_experiments/). So it would be great if they upload at least DEL benchmark results for few samples.

Our approach in providing the steps for reproducing our analyses was to, as much as possible, start from the source public data. For this reason, the instructions in the README of 1_data_preparation should be executed before attempting to reproduce any other figure.

In this particular case, this means downloading the HGSV2 dataset and generating the benchmark datasets for each individual sample:

wget https://ftp.1000genomes.ebi.ac.uk/vol1/ftp/data_collections/HGSVC2/release/v2.0/integrated_callset/variants_freeze4_sv_insdel_alt.vcf.gz

(This downloads the HGSVC2 dataset)

(It is now necessary to divide benchmark insertions from tandem duplications, instructions are reported in https://github.com/kensung-lab/survindel2_paper_experiments/blob/main/1_data_preparation/README.txt)

```
mkdir by-sample
bcftools query -l variants_freeze4_sv_insdel_alt.vcf.gz | while read sample ; do bcftools view variants_freeze4_sv_insdel_alt.vcf.gz -i "SVTYPE=='DEL'" -s $sample --min-ac=1 -Oz -o by-sample/$sample.DEL.vcf.gz ; done
bcftools query -l variants_freeze4_sv_insdel_alt.vcf.gz | while read sample ; do bcftools view variants_freeze4_sv_insdel_alt.DUP.vcf.gz -i "SVTYPE!='DEL'" -s $sample --min-ac=1 -Oz -o by-sample/$sample.DUP.vcf.gz ; done
bcftools query -l variants_freeze4_sv_insdel_alt.vcf.gz | while read sample ; do bcftools view variants_freeze4_sv_insdel_alt.vcf.gz -i "SVTYPE!='DEL'" -s $sample --min-ac=1 -Oz -o by-sample/$sample.INS.vcf.gz ; done
```

(This, barring unforeseen failures, generates benchmark datasets for each individual, including 1_data_preparation/by-sample/HG00512.DEL.vcf.gz.).

We find this approach more transparent, as it makes clear how we obtained the benchmark datasets, and that we did not filter/manipulate them unfairly.

It should be noted that there are also instances in which obtaining some of the results requires significant computation time/resources. In those cases, we also provided pre-computed results.

2. For HG002 with hg19 reference calls, truvari failed with error "ERROR:root:No SAMPLE columns found in vcf", though it worked for Manta and DRAGEN vcfs.

I think it would be great if authors could show that the performance of their tool is better than Manta/DRAGEN (and others) by running Truvari on the DEL variant types.

We have included instructions to https://github.com/kensung-lab/survindel2_paper_experiments/blob/main/hg002-additional/README.txt on how to use truvari to benchmark the HG002 on hg19 calls, on both the GIAB and the Sniffles2 benchmark.

We did not encounter the error reported, but truvari failed on the GIAB benchmark because it was missing the contigs lengths (we show how to work around it in the README).

Here are the results:

Sniffles benchmark:

DEL:

Manta recall 0.39
Manta precision 0.80

SurVIndel2 recall 0.51
SurVIndel2 precision 0.76

DUP:

Manta recall: 0.15
Manta precision: 0.90

SurVIndel2 recall 0.40
SurVIndel2 precision 0.89

GIAB benchmark:

DEL:

Manta recall 0.52
Manta precision 0.79

SurVIndel2 recall 0.73
SurVIndel2 precision 0.90

DUP:

Manta recall 0.22
Manta precision 0.86

SurVIndel2 recall 0.50
SurVIndel2 precision 0.88

There are a couple of points worth mentioning. We have previously explained why we disable the comparison of inserted sequences in duplications when using truvari: truvari assumes that a tandem duplication duplicates the reference sequence exactly once, whereas SurVClusterer does not have this assumption.

This way, the precisions for duplications become very high. We briefly checked the Manta tandem duplications reported as true positives by truvari on the Sniffles benchmark. Here are the first two:

```
chr1 1584516 MantaDUP:TANDEM:61:2:6:0:0:0 A <DUP:TANDEM> 289 PASS
END=1651190;SVTYPE=DUP;SVLEN=66674;IMPRECISE;CIPOS=-578,579;CIEND=-437,437;Pct
SeqSimilarity=0;PctSizeSimilarity=0.0587;PctRecOverlap=0.0297;SizeDiff=-62761;StartDistance=
26;EndDistance=-66648;TruScore=2;MatchId=32.1.0 GT:FT:GQ:PL:PR
0/1:PASS:289:339,0,761:53,26
chr1 1588374 MantaDUP:TANDEM:52:0:1:1:0:0 C <DUP:TANDEM> 23 PASS
END=1653752;SVTYPE=DUP;SVLEN=65378;IMPRECISE;CIPOS=-297,297;CIEND=-679,680;Pct
SeqSimilarity=0;PctSizeSimilarity=0.0008;PctRecOverlap=0.0008;SizeDiff=-65327;StartDistance=
61112;EndDistance=-4266;GTMatch;TruScore=0;MatchId=33.0.0 GT:FT:GQ:PL:PR
0/1:PASS:23:73,0,999:133,25
```

For the first, Sniffles2 reported two insertions near the first breakpoint, one of size 681 and the other 3,913. The tandem duplication of Manta could not have generated such insertion, since it reports a duplication of 66,674 bps.

SurVClusterer takes this into account: it allows the duplicated region to repeat multiple times, if necessary, in order to match the inserted sequence; however, if the duplicated region is larger than the inserted sequence, it will not match the two.

We are very puzzled by the second one, since Sniffles2 does not report any insertion close to either breakpoints. We do not know why Truvari would consider this duplication as a true positive.

Second, in the Sniffles benchmark, it seems that truvari reports a lower precision for both Manta and SurVIndel2 (SurVClusterer reports nearly 0.9).

When manually checking the SurVIndel2 false positives according to truvari, however, many of them are clearly supported by long reads. We have uploaded the IGV screenshots for all false positives to the GitHub repository. We are not sure why truvari reports many of them as false positives.

Sincerely,

Ramesh Rajaby and Wing-Kin Sung.

REVIEWER COMMENTS

Reviewer #3 (Remarks to the Author):

I would like to thank the authors for the detailed response to my earlier suggestions. I understand that the authors have put a README file with all the commands that were used in the validation. In the earlier suggestion, I mentioned that some of the files were not available on Github repo. I totally agree with the authors that some executions may need significant time/resources. Therefore, it would be great if authors make only these three files available on their GitHub repo (or send the link to these files if they are already available)

HG00512

1. HG00512.DEL.vcf.gz
2. HG00512.survindel2.ml.DEL.alt.vcf.gz

HG002

1. HG002.survindel2.ml.DEL.alt.vcf.gz

It is also very interesting to know that truvari assumes tandem duplications are duplicated in the reference sequence exactly once. So it means that any benchmarking for tandem duplications using truvari were not accurate. The authors of Truvari have recently released a tandem-repeat benchmark (<https://www.nature.com/articles/s41587-024-02225-z>) that also used Truvari for validation, I am curious if they also used the same concept i.e. duplicated once in reference. It would be worth checking with Truvari developers or reporting these issues as this tool has become a standard tool for any validations.

We thank the reviewers for the kind feedback. We will reply to their reviews point by point:

Reviewer 3

I would like to thank the authors for the detailed response to my earlier suggestions. I understand that the authors have put a README file with all the commands that were used in the validation. In the earlier suggestion, I mentioned that some of the files were not available on Github repo. I totally agree with the authors that some executions may need significant time/resources. Therefore, it would be great if authors make only these three files available on their GitHub repo (or send the link to these files if they are already available)

As suggested by the reviewer, we have uploaded the three requested files, in:

https://github.com/kensung-lab/survindel2_paper_experiments/tree/main/1_data_preparation/by-sample

https://github.com/kensung-lab/survindel2_paper_experiments/tree/main/truvari

It is also very interesting to know that truvari assumes tandem duplications are duplicated in the reference sequence exactly once. So it means that any benchmarking for tandem duplications using truvari were not accurate. The authors of Truvari have recently released a tandem-repeat benchmark (<https://www.nature.com/articles/s41587-024-02225-z>) that also used Truvari for validation, I am curious if they also used the same concept i.e. duplicated once in reference. It would be worth checking with Truvari developers or reporting these issues as this tool has become a standard tool for any validations.

Thank you for recommending the paper, which we were unaware of.

Upon reading it, it seems that Truvari now has a new comparison algorithm, called refine, which improves over the previous one in two ways:

- 1) A problem in comparing indels (whether small or large) in TR regions is that the indel can shift. For example, we can have two deletions, one deleting the first copy and one deleting the last copy of the same TR, and while they are disjoint, they should be recognized as producing the same haplotype. They improved the way in which this work. This problem was one of the reasons that pushed us to develop an in-house comparison algorithm.
- 2) An indel can not only shift, but also be “decomposed” (for lack of a better term) into multiple smaller indels. For example, if a region AA becomes AAAA, we can have a single insertion of AA, or two distinct insertions of A, and we are effectively representing the same indel. The authors have an algorithm that seems to be able to correctly compare indels under these conditions. This is a difficult problem that does occur in practice (especially with noisy long reads).

However, none of these changes appear to improve the INS vs DUP comparison, which tends to be a problem mostly when comparing a long reads-derived benchmark dataset to a short reads-derived called dataset.

Furthermore, it is not immediately clear to us how the refine command in Truvari directly translates to comparing sets of SVs. From its documentation (<https://github.com/ACEnglish/truvari/wiki/refine>):

The regions spanned by subset.bed should be shorter and focused around the breakpoints of putative FNs/FPs. Haplotypes from these boundaries are fed into a realignment procedure which can take an extremely long time on e.g entire chromosomes.

However, for example, Manta calls several large deletions and duplications, often spanning almost a whole chromosome. Furthermore, they add:

Also, the genotypes within these regions must be phased.

However, this is not generally possible when using short reads.

We will try and get in touch with the authors of Truvari and see if we missed something and if there is indeed a way to extend the comparison algorithm of INS vs DUP.

Sincerely,

Ramesh Rajaby and Wing-Kin Sung.

REVIEWER COMMENTS

Reviewer #3 (Remarks to the Author):

I would like to thank the reviewers for providing the following VCF files for HG00512 and HG002 sample

HG00512

1. HG00512.DEL.vcf.gz
2. HG00512.survindel2.ml.DEL.alt.vcf.gz

HG002

1. HG002.survindel2.ml.DEL.alt.vcf.gz

The Truvari (v4.1) run on HG00512 (using the above two DEL vcf files) with the same parameters that the authors provided in the README file produced the following results. It would be great if the authors could compare this with their results and update them.

```
"precision": 0.8440798882173747,  
"recall": 0.610386228126738,  
"f1": 0.7084589093098503,
```

Regarding the comparison of DEL variants between SurVIndel2 (using the above DEL vcf) and DRAGEN based on Truvari (v4.1) produced the following results.

DRAGEN

```
"precision": 0.830099350472806,  
"recall": 0.6691725891079381,  
"f1": 0.7409993036060416,
```

SurVIndel2

```
"precision": 0.810636420815687,  
"recall": 0.6229756676916839,  
"f1": 0.7045235870445897,
```

So the Truvari-based analysis shows DRAGEN performs better than SurVIndel2 for DEL variants.

Although the authors have explained a few times the issues of Truvari based comparison and why the in-house evaluation tool is better than Truvari, I would still recommend providing all the results based on Truvari. Also, it is recommended to provide the valid reasons in the manuscript of using in-house tool against the standard evaluation tool.

We thank the reviewers for the feedback. We will reply to their reviews point by point:

Reviewer 3

I would like to thank the reviewers for providing the following VCF files for HG00512 and HG002 sample

HG00512

1. HG00512.DEL.vcf.gz
2. HG00512.survindel2.ml.DEL.alt.vcf.gz

HG002

1. HG002.survindel2.ml.DEL.alt.vcf.gz

The Truvari (v4.1) run on HG00512 (using the above two DEL vcf files) with the same parameters that the authors provided in the README file produced the following results. It would be great if the authors could compare this with their results and update them.

```
"precision": 0.8440798882173747,  
"recall": 0.610386228126738,  
"f1": 0.7084589093098503,
```

Regarding the comparison of DEL variants between SurVIndel2 (using the above DEL vcf) and DRAGEN based on Truvari (v4.1) produced the following results.

DRAGEN

```
"precision": 0.830099350472806,  
"recall": 0.6691725891079381,  
"f1": 0.7409993036060416,
```

SurVIndel2

```
"precision": 0.810636420815687,  
"recall": 0.6229756676916839,  
"f1": 0.7045235870445897,
```

So the Truvari-based analysis shows DRAGEN performs better than SurVIndel2 for DEL variants.

Although the authors have explained a few times the issues of Truvari based comparison and why the in-house evaluation tool is better than Truvari, I would still recommend providing all the results based on Truvari. Also, it is recommended to provide the valid reasons in the manuscript of using in-house tool against the standard evaluation tool.

As recommended, we modified our supplementary section “Comparing deletions and tandem duplications” to include why we use the in-house tool rather than Truvari. Furthermore, we have added two panels to Fig. S8, that compare Manta, DRAGEN and SurVIndel2 using Truvari.

We have also modified the scripts in the truvari subfolder to make them work to Truvari 4+. We also noticed that when comparing Manta and DRAGEN, Truvari would give the following warning:

```
[WARNING] Unresolved SVs (e.g. ALT=<DEL>) are filtered when `--pctseq != 0`
```

And many calls would be ignored. Therefore, we now compare deletions with `--pctseq 0`.

Regarding the comparison between SurVIndel2 and DRAGEN, we get very different numbers.

We have uploaded the DRAGEN deletions we have obtained for HG00512 to https://github.com/kensung-lab/survindel2_paper_experiments/tree/main/dragen (along with the script to obtain them, in README.txt).

DRAGEN

```
"precision": 0.8632424434580427,  
"recall": 0.44937403909510215,  
"f1": 0.5910618199468787,
```

SurVIndel2

```
"precision": 0.8249924173491052,  
"recall": 0.5734680430485394,  
"f1": 0.6766108881942217,
```

(https://github.com/kensung-lab/survindel2_paper_experiments/blob/main/truvari/README.txt contains the command we used)

DRAGEN reports 4732 deletions for HG00512, while the benchmark contains 9106 deletions. Therefore, it is unlikely to achieve a sensitivity higher than $4732/9106 = 0.52$, regardless of the comparison method.

In case we are using different datasets or we have made a mistake in downloading the DRAGEN results, please let us know.

REVIEWERS' COMMENTS

Reviewer #3 (Remarks to the Author):

I would like to thank the authors for their satisfactory responses. There are no other concerns about the excellent work the authors did in this paper.

I have a last suggestion for authors to cross check the HG002 DEL comparison between DRAGEN and SurVIndel2 using truvari and report the correct outputs in the final version. The following results that was mentioned in the last review could belong to HG002 dataset (not HG00512).

DRAGEN

"precision": 0.830099350472806,
"recall": 0.6691725891079381,
"f1": 0.7409993036060416,

SurVIndel2

"precision": 0.810636420815687,
"recall": 0.6229756676916839,
"f1": 0.7045235870445897,

We thank the reviewers for the feedback. We will reply to their reviews point by point:

Reviewer 3

I have a last suggestion for authors to cross check the HG002 DEL comparison between DRAGEN and SurVIndel2 using truvari and report the correct outputs in the final version. The following results that was mentioned in the last review could belong to HG002 dataset (not HG00512).

DRAGEN

```
"precision": 0.830099350472806,  
"recall": 0.6691725891079381,  
"f1": 0.7409993036060416,  
SurVIndel2
```

```
"precision": 0.810636420815687,  
"recall": 0.6229756676916839,  
"f1": 0.7045235870445897,
```

We could not replicate the numbers, using Truvari 4.1, even for SurVIndel2. We obtain

```
"precision": 0.7815208275090858,  
"recall": 0.5774434462604178,  
"f1": 0.6641588578474773,
```

While for the dataset provided by the reviewer for DRAGEN on HG002 (<https://www.ncbi.nlm.nih.gov/pmc/articles/PMC10802302/>) we obtained the following numbers, again with Truvari 4.1

```
"precision": 0.800033892560583,  
"recall": 0.5116354583829419,  
"f1": 0.624129407381033,
```

It should be noted that we ran SurVIndel2 on a 50x HiSeq dataset, while the DRAGEN dataset was 35x NovaSeq 6000, so the comparison is not completely fair.

This brings us to the main reason why we think a comparison for HG002 would not be meaningful. A meaningful comparison requires, in our opinion, that the same input data is used. This was possible for the 34 samples in the HGSVC2 benchmark because the same set of reads, sequenced by the New York Genome Institute, were used to run SurVIndel2 and DRAGEN. However, for HG002, we selected a subset of Illumina reads (50x depth in total) from the GIAB project (which is 300x in total), and ran the different callers. We could not find a DRAGEN callset from the same set of reads.

To summarise, we would like not to include the comparison in the paper, because the comparison would not be not fair, as different coverage, sequencing platform, laboratories, etc. could influence the results. Furthermore, it would add little to the paper since

- 1) We have already compared SurVIndel2 to DRAGEN in 34 samples where a fair comparison is possible
- 2) DRAGEN is not a direct competitor of SurVIndel2, as it is a commercial product, while SurVIndel2 is open source and freely downloadable/usable
- 3) Even disregarding the above mentioned points, in the above mentioned test, SurVIndel2 still outperforms DRAGEN on HG002 (we would like to stress once again that it is not a completely fair comparison, and thus we do not think it should be reported in the paper)

Ramesh Rajaby and Wing-Kin Sung